# Overcoming small-bandgap charge recombination in visible and NIR-light-driven hydrogen evolution by engineering the polymer photocatalyst structure

Mohamed Hammad Elsayed [1,2,3,4], Mohamed Abdellah[5,6,7],
Ahmed Zaki Alhakemy[8], Islam M. A. Mekhemer[4], Ahmed Esmail A. Aboubakr[9,10,11],
Bo-Han Chen[12], Amr Sabbah [3,13], Kun-Han Lin [4], Wen-Sheng Chiu[14],
Sheng-Jie Lin[14], Che-Yi Chu[14], Chih-Hsuan Lu[12], Shang-Da Yang [12],
Mohamed Gamal Mohamed[15], Shiao-Wei Kuo[15], Chen-Hsiung Hung[11],
Li-Chyong Chen [3,13,16], Kuei-Hsien Chen [2,3] & Ho-Hsiu Chou [4]✉

Designing an organic polymer photocatalyst for efficient hydrogen evolution with visible and near-infrared (NIR) light activity is still a major challenge. Unlike the common behavior of gradually increasing the charge recombination while shrinking the bandgap, we present here a series of polymer nanoparticles (Pdots) based on ITIC and BTIC units with different π-linkers between the acceptor-donor-acceptor (A-D-A) repeated moieties of the polymer. These polymers act as an efficient single polymer photocatalyst for H2 evolution under both visible and NIR light, without combining or hybridizing with other materials. Importantly, the difluorothiophene (ThF) π-linker facilitates the charge transfer between acceptors of different repeated moieties (A-D-A-(π-Linker)-A-D-A), leading to the enhancement of charge separation between D and A. As a result, the PITIC-ThF Pdots exhibit superior hydrogen evolution rates of 279 μmol/h and 20.5 μmol/h with visible (>420 nm) and NIR (>780 nm) light irradiation, respectively. Furthermore, PITIC-ThF Pdots exhibit a promising apparent quantum yield (AQY) at 700 nm (4.76%).

Conjugated organic polymers have attracted attention as a new photocatalytic approach for hydrogen production due to their tunable bandgap and notable ability to modify the molecular structure[1–5]. In an ideal scenario, a photocatalyst must be a semiconductor material that possesses a band gap energy equal to or greater than the electrochemical potential (1.23 eV) necessary for propelling water-splitting reactions. However, for practical purposes, the band gap of a photocatalyst must be in the range of 1.8 to 2.0 eV[6]. Nevertheless, this requirement alone is insufficient for enabling overall water-splitting reactions since different combinations of conduction band (CB) and valence band (VB) can result in the same band gap energy. To fulfill the

criteria, the CB must have a more negative potential than that of the proton reduction reaction, while the VB must have a more positive potential than that of the water oxidation reaction[7–12]. The demanding specifications of the photocatalyst material, therefore, pose significant limitations and challenges in the selection of suitable materials for overall water-splitting reactions[11,12]. On the other hand, numerous studies have reported the effective utilization of a sacrificial hole scavenger to enhance the photocatalytic production of hydrogen from water. The hydrophobic nature of the majority of conjugated polymers counts as a problem that minimizes their water dispersity[13–15]. Therefore, some amphiphilic surfactants are utilized to convert the bulk

polymer into polymer nanoparticles (Pdots), which enhance their water dispersity, increase the active area, and reduce the diffusion length of the charge carrier[16–19]. Numerous series of conjugated polymers have been reported as photocatalysts for hydrogen production under UV and visible light irradiation, excluding the near-infrared (NIR) region. It is worth noting that the NIR region accounts for more than 50% of the solar radiation spectrum[20–26]. Full harvesting of solar light, spanning from the visible to the near-infrared (NIR) region, is always the primary goal and a significant challenge for photocatalysts. This challenge is particularly pronounced for the approach of conjugated polymer photocatalysts, which face obstacles such as insufficient photocatalytic activity of narrow bandgap semiconductors, direct conversion of NIR light energy into heat, or the low photon energy of NIR light[27]. Unfortunately, only a few photocatalysts demonstrate NIR activity for $H_2$ evolution, and they are mostly limited to composite, heterostructured, or hybridized materials[27–31]. Moreover, previous studies on organic polymer photocatalysts have not presented a polymer photocatalyst with NIR activity for $H_2$ evolution.

Small molecules of organic semiconductors, namely ITIC and BTIC[32,33], have been widely used in organic solar cell applications due to their narrow bandgap, efficient light absorption in the visible and NIR regions, high charge mobility, and easily tunable energy levels[34–36]. The ITIC structure consists of a bulky fused aromatic ring as a core donor (D) with two terminal acceptors (A) of 1,1-dicyanomethylene-3-indanone, which is constructed as A-D-A, and ITIC stands for 2,2′-[[6,6,12,12-tetrakis(4-hexylphenyl)−6,12-dihydrodithieno[2,3-d:2′,3-d′]-s-indaceno[1,2-b:5,6-b′]dithiophene-2,8-diyl]-bis[methylidyne(3-oxo-1H-indene-2,1(3H)-diylidene)]]bis(propanedinitrile)[37]. BTIC is constructed as A-DAD-A and stands for (2,2′-((2Z,2′Z)-((12,13-bis(2-ethylhexyl)−3,9-diundecyl-12,13-dihydro-[1,2,5]thiadiazolo[3,4-e] thieno[2″,3″:4′,5′] thieno [2′,3′:4,5] pyrrolo [3,2 g] thieno [2′,3′:4,5] thieno [3,2-b] indole-2,10-diyl) bis(methanylidene)) bis(3-oxo-2,3-dihydro-1H-indene-2,1-diylidene))dimalononitrile)[38]. Recently, Tian's group, Cooper's group, and McCulloch's group integrated ITIC or BTIC small molecules as acceptors with other donor polymers to construct hybrid photocatalysts for hydrogen evolution[14,16,39]. The remarkable ability to modify the molecular structure of organic materials allows us to create a series of ITIC- and BTIC-based polymers capable of absorbing a broad range of the solar spectrum, including the visible and NIR regions, while avoiding charge recombination.

In this study, we present a series of ITIC- and BTIC-based polymeric photocatalysts for H2 evolution under visible and NIR light, without the need for combination or hybridization with other materials. Our molecular design involves introducing and tuning the π-linker between the repeated ITIC or BTIC moieties of the polymer, resulting in polymer structures of A-D-A-(π-linker)-A-D-A and A-DAD-A-(π-linker)-A-DAD-A, respectively (Fig. 1). We use three different π-linkers (X), namely phenyl (Ph), thiophene (Th), and 3,4-difluorothiophene (ThF) units, as comonomers on the terminal acceptor ring for polymerization using cross-coupling methods (Fig. 1). The choice of different π-linker groups leads to structures with varying planarity, which affects the conjugation and charge transfer between acceptors of different repeated moieties in the polymer. This flexibility allows us to tune the bandgap, absorption properties, and charge separation between D and A during the photocatalytic reaction. Interestingly, our constructed polymer with a ThF π-linker exhibits a redshift in absorption to the NIR region and enhances the charge separation between D and A simultaneously, contrary to the common behavior of

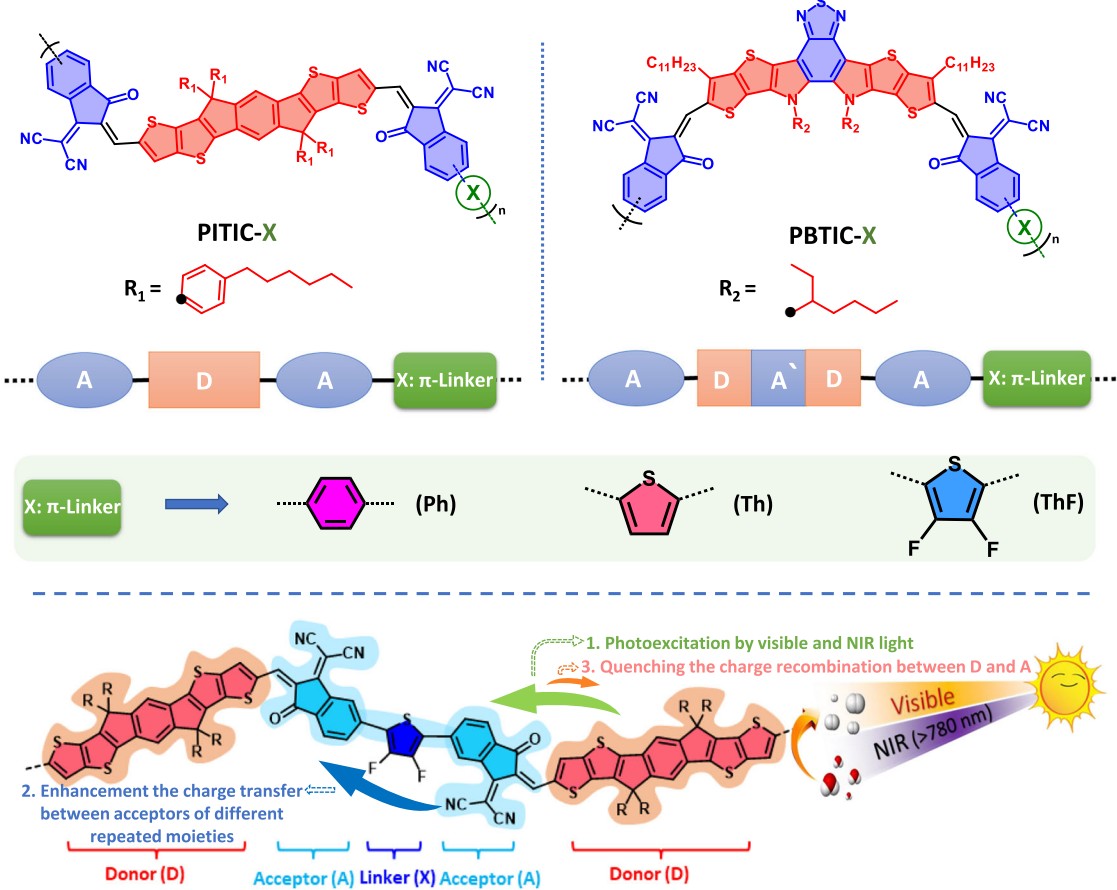

**Fig. 1 | Schematic illustration of design strategy and polymer structures.** Chemical structures of the PITIC-X and PBTIC-X polymers are shown, featuring various π-linker units (X = Ph, Th, and ThF). Additionally, a schematic diagram illustrates the charge transfer through the PITIC-ThF polymer chain during the photocatalytic reaction under visible and NIR light.

increased charge recombination with a shrinking bandgap (Fig. 1). Moreover, the higher crystallinity of the PITIC-based polymer, as well as the charge distribution between A and A' in the PBTIC-based polymer, explains the superior photocatalytic activity of the former over the latter.

## Results

### Polymer and Pdots preparation and characterizations

The synthesis routes of the designed series of PITIC-X-[40] and PBTIC-X-based conjugated polymers are presented in Supplementary Figs. 1–4. We successfully synthesized the three PITIC-X and PBTIC-X-based polymers using Suzuki Miyaura and Stille coupling polymerizations. As a result, we obtained the polymers PITIC-Ph, PITIC-Th, PITIC-ThF, PBTIC-Ph, PBTIC-Th, and PBTIC-ThF, respectively. The detailed procedures for the synthesis are described in the Supporting Information. The as-prepared polymers underwent characterization using various techniques, including $^1$H nuclear magnetic resonance (NMR) spectroscopy, mass spectra (Supplementary Figs. 5–17), thermogravimetric analysis (TGA) (Supplementary Table 1 and Supplementary Fig. 18), Fourier transform infrared (FTIR) spectroscopy (Supplementary Fig. 19a), and X-ray photoelectron spectroscopy (XPS) (Supplementary Figs. 19b, 20, 21, and 22). The residual palladium content was determined by inductively coupled plasma-mass-spectrometry (ICP-MS), and the polymer molecular weight using Gel permeation chromatography (GPC) as shown in Supplementary Tables 3 and 4, respectively. The bulk polymer was converted into polymer nanoparticles (Pdots) using precipitation methods. This involved dispersing the polymer in water without additional surfactants, such as PS-PEG-COOH or Triton, under vigorous sonication. The detailed preparation procedure is described in the Supplementary Fig. 23.

The photographs of all six polymers in THF solutions exhibit distinct colors, indicating that their optical properties can be easily tuned by introducing π-linker units on the acceptor rings (Fig. 2a). The normalized ultraviolet-visible (UV-vis) absorption spectra of the six Pdots in water solutions are presented in Fig. 2b, and the corresponding data are summarized in Supplementary Table 2. The monomer Br-ITIC-Br exhibits strong absorption in the visible range of 550–750 nm, with a peak at 669 nm and a small shoulder at 620 nm (Supplementary Fig. 24). The ITIC polymer prepared using the thiophene π-linker (PITIC-Th) shows a similar absorption onset to Br-ITIC-Br, with a slightly blue-shifted peak at 657 nm. Notably, substituting the thiophene group with a phenyl group (PITIC-Ph) results in a significant blue shift, with a peak at 551 nm. On the other hand, the introduction of the difluorothiophene group (PITIC-ThF) leads to a small new shoulder at 739 nm and a red-shifted absorption onset. This demonstrates that the incorporation of the difluorothiophene group extends the absorption into the near-infrared (NIR) region, which is advantageous for capturing a larger portion of the solar spectrum. The various comonomer π-linker units with PBTIC-X-based polymers exhibit a red-shifted absorption compared to that of PITIC-X-based polymers. This suggests that the A-DAD-A structure possesses greater conjugation than A-D-A, resulting in a more pronounced red shift in the absorption (Fig. 2b). Significantly, the transformation of the bulk polymers (dissolved in THF) into Pdots (dispersed in water) is accompanied by a redshift of 20–50 nm. This shift is attributed to the J aggregation of the polymer chain in an aqueous solution, as depicted in Supplementary Figs. 25a, b[41,42]. Based on the Tauc plots illustrated in Supplementary Figs. 25c, d, it can be observed that the $E_g$ values of these polymers range from 1.45 to 2.05 eV. Furthermore, when dissolved in water, these polymers exhibit a decrease in $E_g$ values by approximately 0.10 eV. This reduction indicates their narrow bandgap nature, enabling efficient absorption across a wide range of visible and near-infrared (NIR) light wavelengths. By employing different π-linkers, the constructed polymers achieve deeper HOMO and LUMO values with the following arrangement of π-linkers ThF > Th > Ph (Fig. 2c). The

aforementioned results demonstrate that the introduction of diverse π-linker units on acceptor rings significantly affects the photophysical properties of the polymers.

The prepared structure and morphology of the Pdots were determined using cryo-transmission electron microscopy (Cryo-TEM), as depicted in Fig. 2d. The six polymers exhibit spherical particles with non-uniform sizes ranging from 30 to 70 nm, similar to those reported in other studies[43,44]. The presence of nanometric spherical particles indicates the formation of Pdots from all the polymers. Furthermore, the sessile drop technique was employed to measure the static contact angles of the as-prepared Pdots at three different locations with water at room temperature. The water contact angles of the PITIC-X polymers were found to be lower than those of the PBTIC-X polymers (Supplementary Table 1 and Supplementary Fig. 27), suggesting a lower hydrophobicity. Although the as-prepared polymers are hydrophobic, their conversion into the Pdot structure enhances their water dispersion and increases the surface area of the photocatalyst. Thus, the conversion of hydrophobic polymers into Pdots, without the need for surfactants, could be advantageous in achieving good water dispersibility and enhancing hydrogen production from water.

### Photocatalytic activity under visible and NIR light

Next, we evaluated the Pdots as photocatalysts for visible light-driven hydrogen evolution. A PAR30 light-emitting diode (LED) lamp with specific specifications (20 W, 6500 K, and λ > 420 nm) was utilized as the light source (refer to Supplementary Fig. 28 for details). To enhance the photocatalytic activity, we incorporated an optimized amount (3%) of the Pt cocatalyst ($H_2PtCl_6$) (refer to Supplementary Fig. 29). Under the optimal conditions and visible light illumination, we recorded the kinetic curve of hydrogen evolution to investigate the efficiency of the photocatalyst (refer to Fig. 3a). From the kinetic curves, we extracted the $H_2$ evolution rate of the six polymers (refer to Fig. 3b). Among them, the PITIC-ThF Pdots exhibited the highest $H_2$ evolution rate (339.7 mmol g$^{-1}$ h$^{-1}$), followed by the PITIC-Th Pdots (168.7 mmol g$^{-1}$ h$^{-1}$) and PITIC-Ph Pdots (106.2 mmol g$^{-1}$ h$^{-1}$). A similar trend was observed for the PBTIC-X series, with the order of $H_2$ evolution being PBTIC-ThF Pdots (269.4 mmol g$^{-1}$ h$^{-1}$) > PBTIC-Th Pdots (121.1 mmol g$^{-1}$ h$^{-1}$) > PBTIC-Ph Pdots (69.8 mmol g$^{-1}$ h$^{-1}$). The $H_2$ evolution rate increased with increasing photocatalyst amount, with a loading of 0.1 mg to 5 mg resulting in an $H_2$ evolution increase from 34 to 279 μmol/h and 27 to 178 μmol/h for the PITIC-ThF Pdots and PBTIC-ThF Pdots, respectively (refer to Fig. 3c). Our photocatalyst demonstrates a significant HER value, suggesting its potential as an effective catalyst for hydrogen production when exposed to visible and NIR light. Additionally, we investigated the photocatalytic activity of the PITIC-ThF Pdots and PBTIC-ThF Pdots under NIR light using a Xenon lamp (AM1.5 and 3000 W m$^{-2}$) with a cutoff filter (λ > 780 nm) as the light source. Over a 4-h period of NIR light irradiation, the PITIC-ThF Pdots generated approximately 11450 ± 800 μmol/g of $H_2$, while the PBTIC-ThF Pdots produced 1715 ± 320 μmol/g (refer to Fig. 3d). Among the two polymers with the ThF π-linker, the PITIC-ThF Pdots demonstrated an $H_2$ evolution rate of 4045 ± 430 μmol h$^{-1}$g$^{-1}$ (20.5 μmol h$^{-1}$), which is approximately more than 5-fold higher than that of the PBTIC-ThF Pdots (708 ± 210 μmol h$^{-1}$g$^{-1}$) (3.54 μmol h$^{-1}$) (refer to Fig. 3e). The HER of the PITIC-ThF Pdots under NIR light exhibits a promising and unprecedented efficiency for a single polymer. Furthermore, the apparent quantum yields (AQYs) of the most efficient PITIC-ThF and PBTIC-ThF Pdots were determined under standard photocatalytic conditions using a light source with a bandpass filter (λ = 420, 500, 550, 600, and 700 nm). As depicted in Fig. 3d, the estimated AQYs for the PITIC-ThF (PBTIC-ThF) Pdots are 3.9 (2.9), 3.2 (2.7), 3.1 (2.5), 3.9 (2.8), and 4.7% (3.1%) at 420, 500, 550, 600, and 700 nm, respectively. These AQY values are consistent with the absorption spectrum of the polymer photocatalyst, with a higher value observed at a longer wavelength of 700 nm. This suggests that these Pdots exhibit good

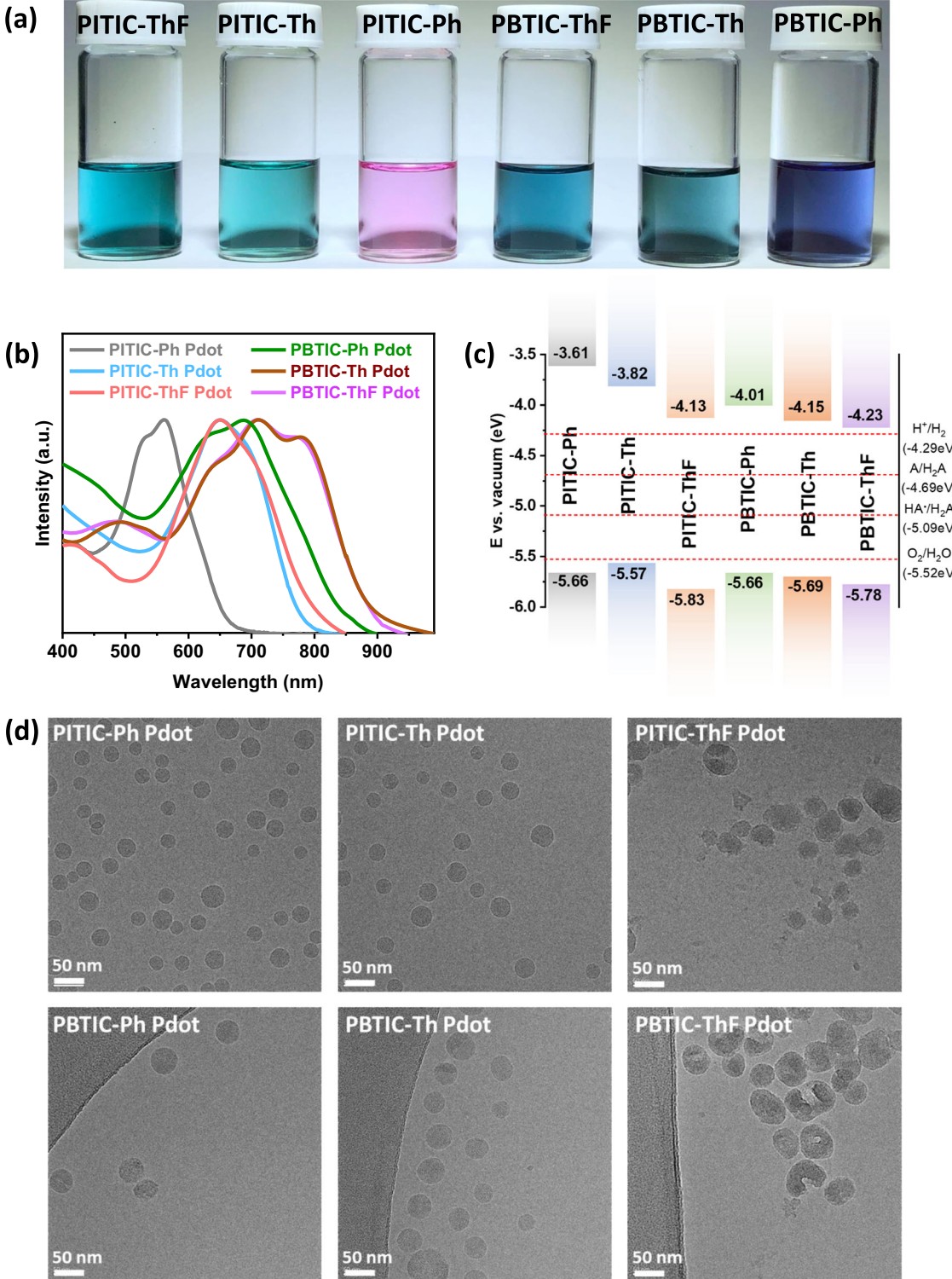

**Fig. 2 | Optical properties and morphology of polymers. a** Photographs depict the six polymers in THF solutions, **b** UV-Vis absorption spectra display the characteristics of the all-polymer dots in water solutions, **c** Energy level diagrams of all the polymers were determined using a photoelectron spectrometer. The dashed lines in the diagram represent the proton reduction potential ($H^+/H_2$), water oxidation potential ($O_2/H_2O$), and the calculated potential of the two-hole ($A/H_2A$) and one-hole ($HA\cdot/H_2A$) oxidation of ascorbic acid at pH 2.6 (the experimentally measured pH of a 0.1 M ascorbic acid solution)[4]. All energy levels and electrochemical potentials are referenced relative to vacuum, with −4.44 V versus vacuum considered equivalent to 0 V versus SHE (Standard Hydrogen Electrode) with respect to the electrochemical potential scale[16]. **d** Cryo-TEM images showcase the PITIC-Ph Pdots, PITIC-Th Pdots, PITIC-ThF Pdots, PBTIC-Ph Pdots, PBTIC-Th Pdots, and PBTIC-ThF Pdots.

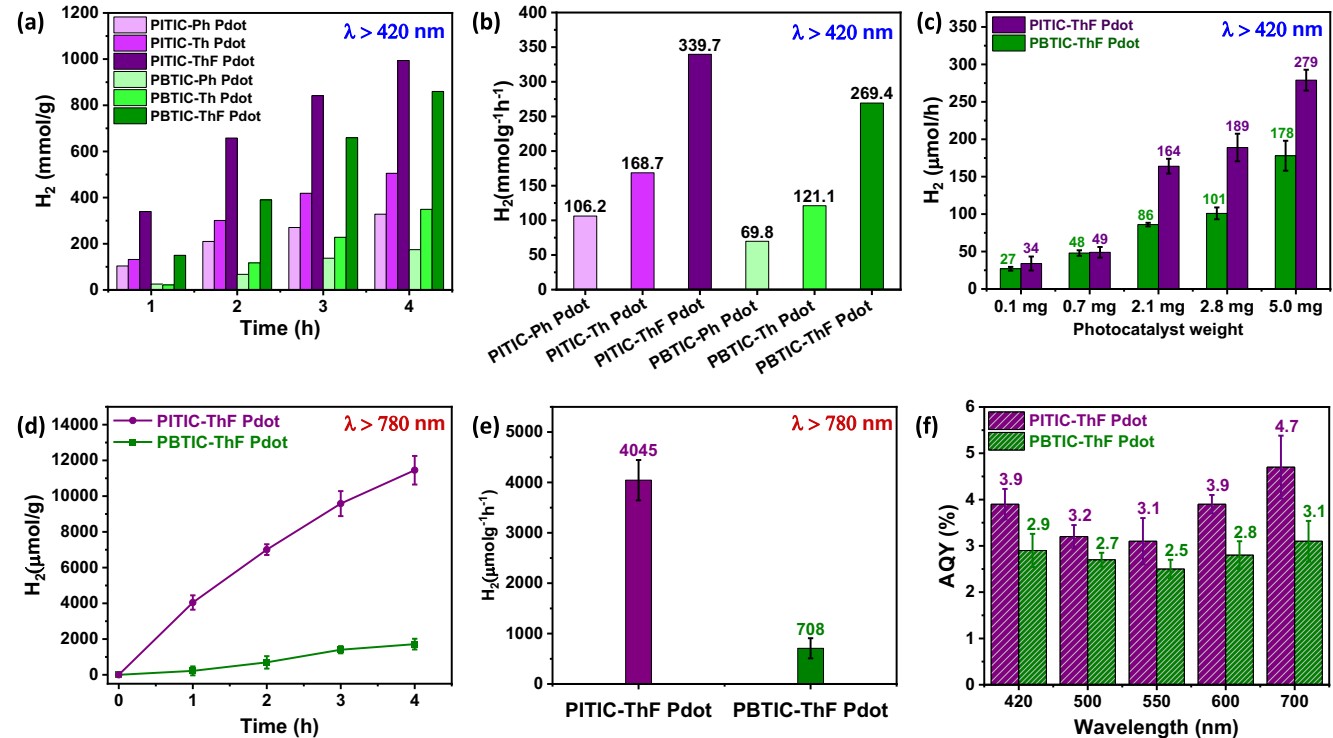

**Fig. 3 | Photocatalytic hydrogen evolution experiments in solution. a** Time course of $H_2$ production for the six Pdots, **b** HER values for the six Pdots, **c** HER values of the PITIC-ThF Pdots and PBTIC-ThF Pdots with different weights of photocatalyst. Experimental conditions include 0.1 M ascorbic acid (AA), white LED light ($\lambda > 420$ nm, 20 W, and 6500 K), and 3% $H_2PtCl_6$. **d** The time course of $H_2$ production for the PITIC-ThF and PBTIC-ThF Pdots under NIR light. **e** The HER values of PITIC-ThF and PBTIC-ThF Pdots under NIR light. Experimental conditions include 0.1 M ascorbic acid (AA), 3% $H_2PtCl_6$, and a xenon lamp light source ($\lambda > 780$ nm and 3000 W m$^{-2}$). **f** The apparent quantum yields for the PITIC-ThF Pdots and PBTIC-ThF Pdots at different wavelengths.

photoresponsivity for hydrogen production throughout the entire visible light region. Notably, the HER values under visible and NIR light, as well as the AQY values at 700 nm of the PITIC-ThF Pdots, are among the highest reported in the literature (refer to Supplementary Tables 5 and 6). Under identical experimental conditions with the use of the same light source, we have compared the efficiency of the PITIC-ThF Pdot photocatalyst under both visible and NIR light ranges (as shown in Supplementary Fig. 30). In addition, we have also evaluated the operational stability of the PITIC-ThF Pdot photocatalyst for 20 h, as demonstrated in Supplementary Figs. 31 and 32. While our designated materials demonstrate promising HER and AQY% efficiency with a sacrificial (AA) half reaction under both visible and NIR light, they face significant challenges in achieving overall water splitting. This is because the presented polymers have HOMO levels that are below the water oxidation potential and LUMO levels that are above the hydrogen reduction potential under photocatalytic conditions (pH = 2.6, acidic), as depicted in Supplementary Fig. 33a. Thus, these materials are more likely to succeed in overall water splitting under acidic conditions, which is generally more challenging due to higher overpotential, and protonation of catalysts. In contrast, neutral solutions are typically preferred for water splitting reactions because of their more favorable conditions. However, in a neutral medium, most of our designated polymers have a LUMO level that is lower than the proton reduction reaction, and a HOMO level that is significantly lower than the water oxidation reaction (Supplementary Fig. 33b), making them only suitable for water oxidation. To overcome the challenges of achieving overall water splitting with our designated materials, one possible solution is to hybridize them with other active materials for $H_2$ reduction, as Prof. Sprik and Prof. Cooper's groups have done[8,10], to produce a Z-scheme photocatalytic system that could achieve overall water splitting and NIR activity simultaneously.

## Unveiling the effect of A-D-A structure and different linkers on the activity

We conducted an investigation into two key aspects to gain a better understanding of the relationship between the structure and activity of the PITIC-X- and PBTIC-X-based polymer photocatalysts. These aspects included the impact of different π-linkers on polymer activity and the differentiation between the ITIC and BTIC moieties. To comprehend the influence of various π-linkers on photocatalytic activity, we employed density functional theory (DFT) and transient absorption (TA) spectroscopy. To better understand the effect of linker on the properties of each polymer, we performed computations using density functional theory (DFT), configuration-interaction constrained DFT (CI-CDFT) and time-dependent DFT (TD-DFT). For the computational details, please refer to the Methodology section and the supporting information. First, the dihedral angles between the linker and the acceptor decrease as we shift from using Ph, Th, to ThF as the linker, which holds for both PBTIC and PITIC series (as shown in Fig. 4a). This is because the H-H repulsion present in the benzene linker and the acceptor reduces by replacing it with Th linker. Upon replacing Th with ThF, not only does this substitution decrease H-H repulsion, but it also introduces the F-H attraction, resulting in a further reduction of the dihedral angle (Supplementary Fig. 34)[45–47]. The smaller dihedral angle between the acceptor and π-linker in PITIC-ThF indicates a more planarized structure with enhanced charge carrier mobility and transfer between the acceptors of different repeated moieties. This, in turn, improves the charge separation between the donor and acceptor for each polymer repeated moiety, resulting in an enhanced exciton dissociation yield and improved photocatalytic activity. Conversely, increasing the dihedral angle with Th and Ph π-linkers reduces planarity and charge transfer between the acceptors, leading to increased charge recombination from the donor to

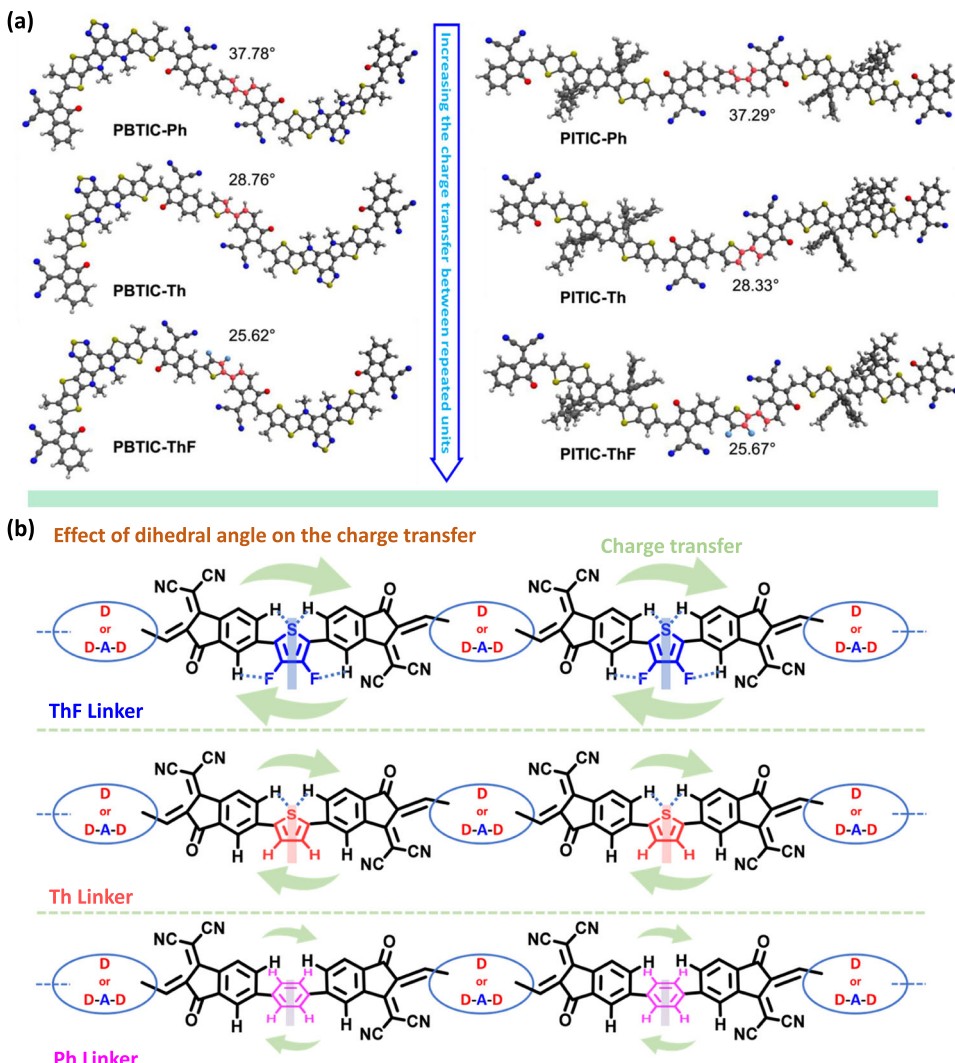

**Fig. 4 | Effect of different likers on the dihedral angle between repeated acceptors. a** The dihedral angle between the linker and the acceptor (indicated in red color) for the dimer of each polymer. **b** Schematic diagram showing the dihedral angle effect of different π-linkers on the charge transfer between acceptors of repeated moieties.

acceptor and consequently less efficient photocatalytic activity (Fig. 4b). As for electronic structure, all dimers exhibit a nearly degenerate HOMO/HOMO-1 and LUMO/LUMO + 1, resulting from the linear combination of HOMOs and LUMOs of the corresponding monomers (as shown in Supplementary Figs. 35 and 36). While all energy values of frontier orbitals remain nearly unchanged across both the PITIC and PBTIC series, it is noteworthy that the molecular orbital distribution extends slightly more into the linker unit in the cases involving Th and ThF for both polymer series. This observation could be partly attributed to the increased planarity found in the polymers containing Th and ThF. To understand how these different structural features affect the electron transport properties, we estimate the electron transfer from an acceptor to another neighboring acceptor. To achieve this, we computed the electronic coupling ($V_{A_1 A_2}$) between these two states using CI CDFT, where the electron transfer rate is proportional to $|V_{A_1 A_2}|^2$. As shown in Supplementary Table 7, the magnitude of $|V_{A_1 A_2}|$ exhibits an ascending trend in the sequence of Ph, Th, and ThF for both series of polymers. Our findings imply that electron transfer along the polymer chain is more favorable when utilizing the ThF linker, which could be attributed to the increased planarity observed in the polymer chain containing ThF. Finally, the excited-state properties of these polymers were investigated using TD-DFT using the aforementioned

dimer model. The vertical $S_1$ energy remains relatively close within each series, with the polymer containing Ph exhibiting a slightly higher $E_{S_1}$ value. While the computational $E_{S_1}$ trend based on the dimer model agree qualitatively with our experiment, it fails to explain a much a larger difference in the optical bandgap (>0.1 eV) between the Ph-containing and Th/ThF-containing polymers. We assumed that the different linker-acceptor dihedral angle distributions of different polymers may be the reason for this discrepancy. Therefore, we conducted a dihedral angle scanning analysis to compute the excited-state energy profile along the linker-acceptor dihedral angle (Supplementary Fig. 37). However, all polymers exhibit very similar characteristics in their energy profiles, which still could not explain the disparities between our computational and experimental findings. We suggest that this distinction may be explained by their different polymer packing behaviors, where the interchain coupling and the dielectric effect can significantly alter the excited-state properties (see Supplementary Fig. 38 and its discussion). Finally, we computed the reorganization energy ($\lambda_{S_1 \to S_0}$) of the $S_1 \to S_0$ transition, where organic semiconductors with small $\lambda_{S_1 \to S_0}$ values are believed to exhibit slower radiationless relaxation and thereby giving higher quantum yield of photoconversion processes. As shown in Supplementary Table 7, the $\lambda_{S_1 \to S_0}$ slightly decreases as we shift from using Ph, Th, to ThF as

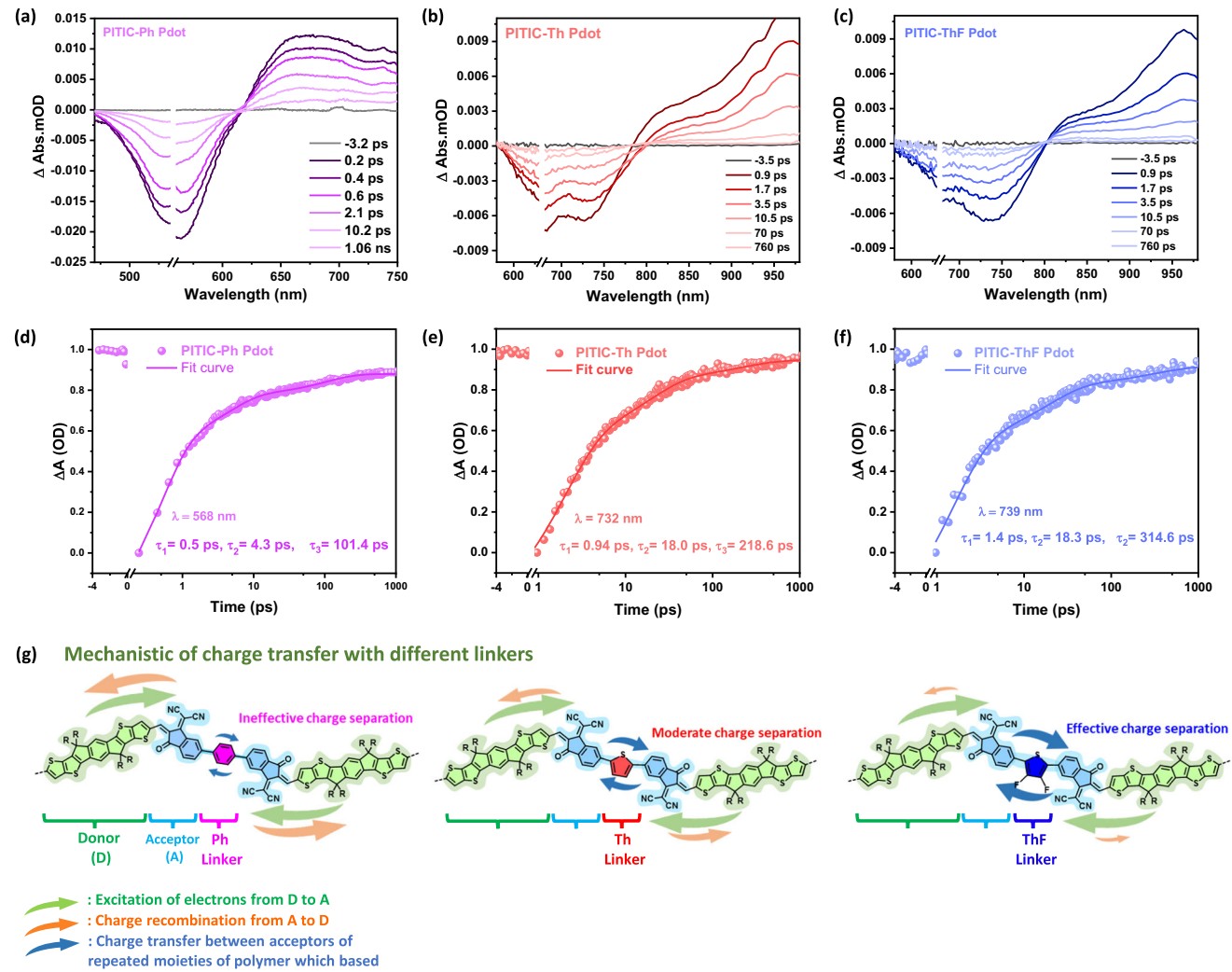

**Fig. 5 | Transient absorption spectroscopy measurements for polymers with different linkers.** Transient absorption spectra of **a** PITIC-Ph Pdots (water solution), **b** PITIC-Th Pdots, and **c** PITIC-ThF Pdots at different delay times. Transient absorption traces of **d** PITIC-Ph Pdot (probed at 568 nm), **e** PITIC-Th Pdots (probed at 732 nm), and **f** PITIC-ThF Pdots (probed at 739 nm). The data were obtained using an excitation wavelength of 550 nm for PITIC-Ph Pdot and 650 nm for both PITIC-Th and PITIC-ThF Pdot, using a power of 10 µW. **g** Schematic diagram presenting the effect of different likers on the charge transfer between acceptors of repeated moieties of the polymer and the charge recombination between the acceptor and donor.

the linker for both polymer series. The trend in $\lambda_{S_1 \rightarrow S_0}$ agrees with the trend of the excited-state lifetime observed in the fs-TAS.

Femtosecond transient absorption spectroscopy (fs-TAS) was used to further study the excited state dynamics of the PITIC-X polymers and the effect of different π-linkers on the excited state lifetime. Figure 5a–c shows the transient absorption spectra of the PITIC-ThF, PITIC-Ph, and PITIC-Th Pdots, respectively. The TA spectra of the three polymers consist of a negative signal assigned to the ground-state bleach of the polymer, which is consistent with the ground-state absorption band, and positive signal, assigned to the singlet exciton absorption of polymer[15,26]. Figure 5d–f shows the bleach recovery dynamics and lifetimes of the three polymers at 568 nm (PITIC-Ph), 732 nm (PITIC-Th), and 739 nm (PITIC-ThF), respectively, which consist of three time components. The results reveal that the PITIC polymers with different π-linkers exhibit distinct bleach recovery dynamics and lifetimes. The recombination of photogenerated charge carriers decelerates more in the following order: ThF > Th > Ph. This can be attributed to the smooth and fast charge transfer between the acceptors of different repeated units in the case of the ThF π-linker with a lower dihedral angle. Moreover, compared to the Th and Ph linkers, the ThF linker in both PITIC and PBTIC polymer series

demonstrates strong quenching emission in the steady-state photoluminescence spectra, highest photocurrent responses in the transient photocurrent response, and smallest arc radii in the electrochemical impedance spectroscopy (EIS) Nyquist plots (Supplementary Fig. 39, with clear discussion).

In Fig. 5g, we present a schematic diagram illustrating the relationship between the different π-linkers and photocatalytic activity based on the previous results. In the case of the Ph π-linker, after photoexcitation, the electrons transfer from D to A, but the large dihedral angle of the Ph π-linker reduces the delocalization of excited electrons between the acceptors of different repeated moieties, leading to the recombination of photogenerated electrons from A to D. This results in a shorter bleach recovery lifetime and ineffective charge separation. With the Th π-linker, the dihedral angle decreases slightly, enhancing the delocalization of excited electrons between acceptors, resulting in a relatively long bleach recovery lifetime accompanied by the inhibition of charge recombination and enhancement of charge separation. On the other hand, the ThF π-linker enables effective charge separation and significantly decreased charge recombination due to its small dihedral angle, which promotes delocalization between acceptors of different repeated moieties. Despite the redshift

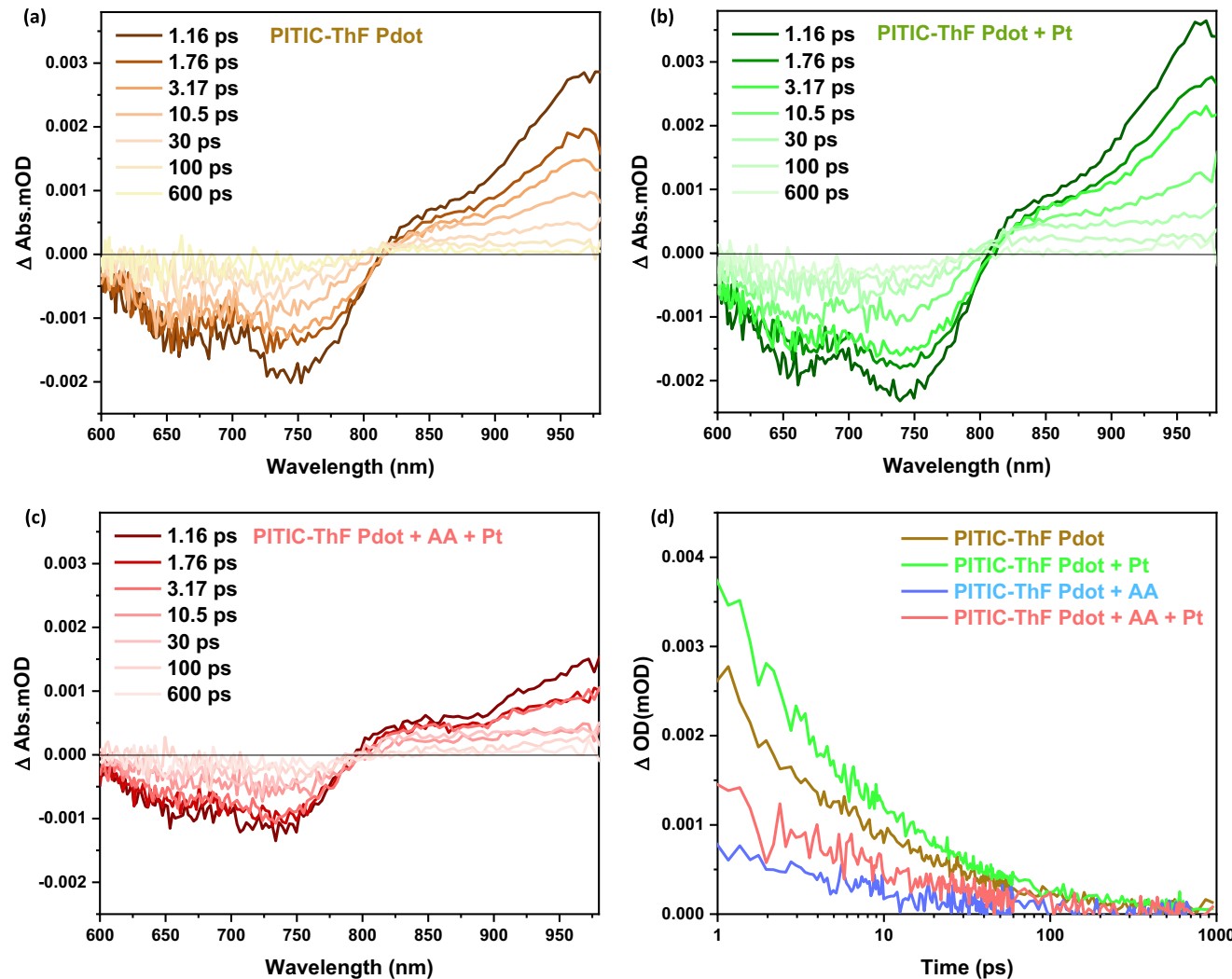

**Fig. 6 | Effect of Pt and AA on the photocatalytic reaction mechanism.**
**a** Transient absorption spectrum for various samples, including pure PITIC-ThF
Pdot, **b** PITIC-ThF Pdot with 3 wt% Pt, and **c** PITIC-ThF Pdot with 0.1 M AA and 3 wt%
Pt, at different time delays after excitation at 650 nm with a power of 0.9 μW. **d** The
decay dynamics of the transient absorption were compared for neat PITIC-ThF
Pdot, PITIC-ThF Pdot + Pt, and PITIC-ThF Pdot + AA + Pt, when excited at 650 nm
and probed at 959 nm, which were assigned to PITIC-ThF Pdot exciton decay.

of the PBTIC-ThF absorption spectra compared to that of PITIC-ThF,
the greater effective photocatalytic activity of PITIC-ThF compared to
PBTIC-ThF under visible and NIR light can be attributed to the higher
crystallinity of PITIC-ThF as well as the charge distribution between A
and A' of the PBTIC-based polymer (Supplementary Fig. 40).

Durrant & McCulloch have reported that the kinetics of charge
recombination in the pure Pdot system differ from those in the full
photocatalytic system, where the presence of a scavenger and Pt
cocatalyst has a significant impact[16]. To gain a better understanding
of the effect of Pt and scavenger on the photocatalytic activity,
transient absorption spectroscopy (TAS) was employed. The TAS
spectra of neat PITIC-ThF Pdot in the wavelength region
(600–980 nm) were recorded after excitation at 650 nm, and the
results were analyzed (Fig. 6a). The TAS spectra of PITIC-ThF Pdot
show a broad ground state bleaching (GSB) between 600 nm and
815 nm, with a maximum peak at 750 nm. Additionally, a photo-
induced absorption (PIA) was observed starting from 815 nm and
reaching a maximum in the NIR region at 959 nm. This PIA is attrib-
uted to the singlet exciton absorption of PITIC-ThF Pdot. In Fig. 6b,
the TAS spectra reveals a larger amplitude with the addition of 3% Pt
to the Pdot solution, which is consistent with suppressed bimolecular
recombination due to electron transfer to Pt. On the other hand, the

addition of AA for Pdot samples with Pt strongly reduced the
amplitude in Fig. 6c, indicating efficient hole extraction in the pho-
tocatalytic system[16]. The exciton decay dynamics of neat PITIC-ThF
Pdot and the photocatalytic system containing Pt or Pt+AA were
compared in Fig. 6e. It was found that the addition of Pt to PITIC-ThF
Pdot resulted in a longer-lived decay transient compared to the neat
PITIC-ThF Pdot, which is consistent with slower bimolecular recom-
bination kinetics with Pt. The further addition of AA resulted in an
accelerated decay of the PITIC-ThF Pdot absorption, consistent with
hole transfer to AA[48,49]. This acceleration was increased even more
with AA alone, indicating that the presence of AA scavenger in this
photocatalytic system is crucial for achieving high photocatalytic
hydrogen evolution rate (Supplementary Fig. 41). The same behavior
was noticed with PITIC-Th Pdot in the presence of Pt or AA+Pt at the
same power energy (Supplementary Fig. 42) and noticed with PITIC-
ThF Pdot at different power energy (Supplementary Fig. 43). These
results provide unequivocal evidence that the presence of AA retards
the recombination of photo-generated charge carriers, leading to a
nearly three-fold increase in recombination time in the presence of
AA and PITIC-ThF Pdot. These results provide insight into the
mechanisms of the photocatalytic system and could be useful for the
design of more efficient photocatalysts.

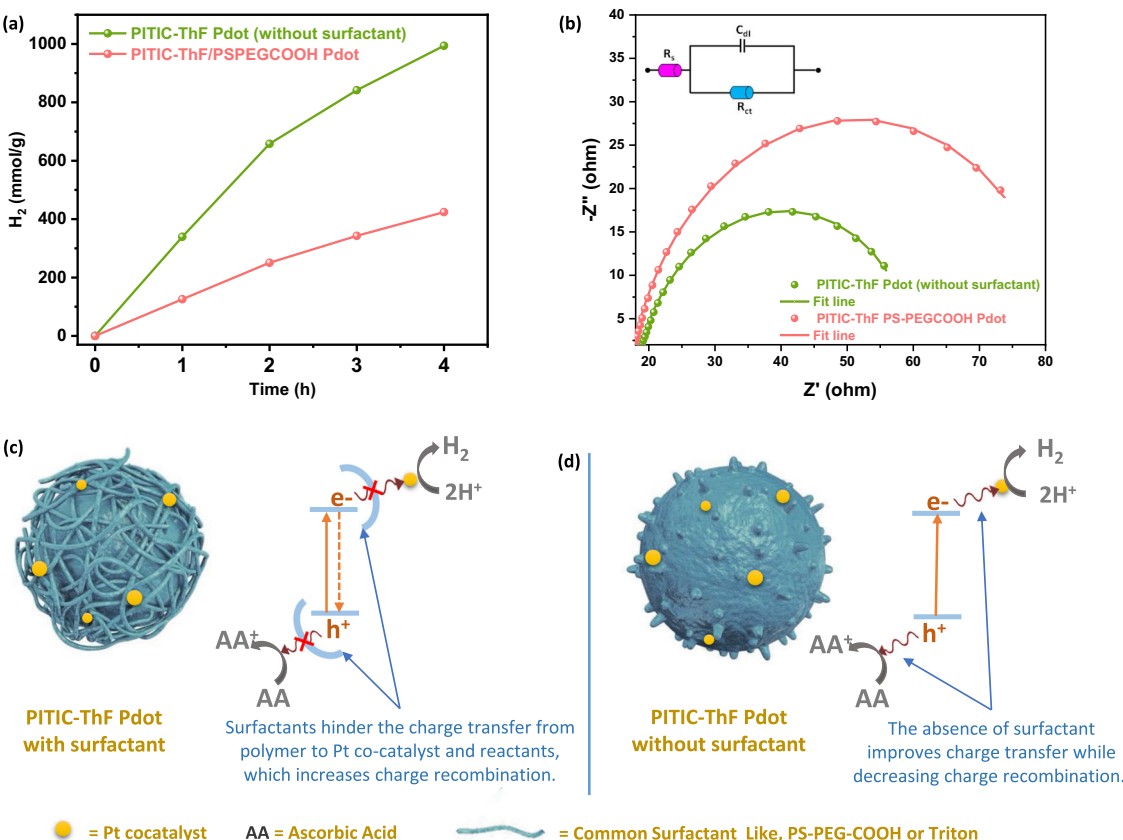

**Fig. 7 | Effect of surfactants used for Pdot preparation on the photocatalytic reaction mechanism. a** Effect of PS-PEGCOOH surfactant on the photocatalytic hydrogen production activity of the PITIC-ThF Pdots. **b** Electrochemical impedance spectroscopy (EIS) of the PITIC-ThF Pdots (with and without a surfactant). **c** Schematic diagram of the PITIC-ThF Pdots with a surfactant. **d** Schematic diagram of the PITIC-ThF Pdots without a surfactant for presenting the effect of the surfactant on the charge transfer between the Pdots and Pt cocatalyst.

## Effect of the free-surfactant Pdot structure on the activity

To clarify the benefits of preparing the Pdot structure without additional surfactants, we studied the effect of common surfactants used in Pdot preparation on the photocatalytic activity of the Pdot photocatalyst. The Pdot structure of PITIC-ThF was prepared using the precipitation method in the absence and presence of common surfactants such as PS-PEG-COOH and Triton, resulting in three types of Pdot structures: PITIC-ThF Pdots, PITIC-ThF/PS-PEG-COOH Pdots, and PITIC-ThF/Triton Pdots. In comparison to the PITIC-ThF Pdots, the PITIC-ThF/PS-PEG-COOH Pdots (Fig. 7a) and PITIC-ThF/Triton Pdots (Supplementary Fig. 45a) inhibit the HER. According to the results obtained from the electrochemical impedance spectroscopy (EIS) technique (Fig. 7b and Supplementary Fig. 45b), the presence of surfactants is found to increase the charge transfer resistance. The EIS Nyquist plot clearly shows that the semicircle diameter for PS-PEG-COOH/PITIC-ThF Pdots is larger than that of PITIC-ThF Pdots, indicating sluggish charge transfer kinetics. Furthermore, the fitting of the Nyquist plot by means of an equivalent circuit model reveals that the PITIC-ThF Pdots exhibit a much lower charge transfer resistance (Rct) of 48.06 Ω compared to PS-PEG-COOH/PITIC-ThF Pdots (Rct = 73.74 Ω) (see Supplementary Table 8). This implies that the absence of surfactants facilitates electron transfer between the polymer electrode and electrolyte solution, while the presence of surfactants increases the resistance and impairs electron transfer. The most plausible explanation for this phenomenon is the non-conductive nature of the surfactant that acts as a barrier when it covers the Pdot particles, thus hindering electron transfer from the semiconducting polymer (PITIC-ThF) to the electrolyte solution[44]. This behavior is also anticipated to occur during the photocatalysis process, where covering PITIC-ThF

with non-conductive surfactants can impede charge transfer from the photocatalyst to the reactants or Pt cocatalyst in the solution. Figure 7d presents a schematic diagram illustrating the charge transfer from the PITIC-ThF Pdots to the Pt cocatalyst and AA. In the absence of PS-PEG-COOH, the excited electrons efficiently transfer to the Pt cocatalyst, leading to effective charge separation and enhanced H₂ evolution activity. However, in the presence of PS-PEG-COOH, the charge transfer from the PITIC-ThF Pdots to the Pt cocatalyst and AA is hindered, resulting in accelerated charge recombination and reduced photocatalytic activity of the Pdot photocatalyst. Therefore, the Pdot preparation method without a surfactant is more efficient in achieving high photocatalytic activity for H₂ evolution.

## Discussion

In summary, different π-linkers (X = Ph, Th, ThF) were utilized as comonomers for the preparation of PITIC and PBTIC-based polymers. These π-linkers successfully altered the absorption spectrum and charge separation during the photocatalytic reaction. Through fs-TAS and DFT calculations, we demonstrated that the various π-linkers significantly affected the molecular planarity and the charge transfer ability between the acceptors of different repeating units in the polymer. This finding allowed for tunability in the optical properties and charge separation between the donor and acceptor during photocatalytic reactions. Consequently, this study presents the first single polymer photocatalyst exhibiting a promising near-infrared (NIR) activity (with >780 nm cut filter) for H2 evolution, achieving a high HER of 20.5 μmol/h and a promising AQY of 4.7% at a wavelength of 700 nm. Additionally, we showed that our surfactant-free method for preparing the Pdot structure exhibited higher photocatalytic activity, as common

surfactants were found to hinder the charge transfer from the Pdot particles to the Pt cocatalyst or ascorbic acid (sacrificial reagent), as evidenced by fs-TAS and electrochemical analysis.

## Methods

### Preparation of PITIC-X and PBTIC-X Pdots

Typically, a certain volume (μL) of the polymer solution (1 mg mL$^{-1}$ in THF) was rapidly poured into water (10 mL) under sonication. Then the solution was purged with N$_2$ (slow rate) on a hot plate at 100 °C for 90 min to remove the THF (Supplementary Fig. 23). To prepare the Pdot structure using surfactants PS-PEG-COOH or Triton, we follow the same method by adding 20% of surfactant with the polymer in the THF solution before being poured to water.

### Photocatalytic H$_2$ evolution measurement

For measuring the H$_2$ evolution under visible light, the Pdot solution prepared with different weights (0.1, 1, 2, 3, and 5 mg/10 mL) containing Ascorbic Acid (AA (0.1 M)) and 3% H$_2$PtCl$_6$ cocatalyst was inserted into the reaction glass container and sealed tightly with a septum. The resulting mixture was degassed by Ar bubbling, prior to illumination. A white light-emitting diode (LED) PAR38 lamp (20 W, 6500 K, Zenaro Lighting; λ > 420 nm) was used as the light source. While, for H$_2$ evolution measurement under NIR light, the Pdot solution (5 mg/10 mL) containing Ascorbic Acid (AA (0.1 M)) and 3% H$_2$PtCl$_6$ cocatalyst illuminated with light source of a Xenon lamp (AM1.5, λ > 780 nm, 3000 W m$^{-2}$). Hydrogen samples were taken with a gas-tight syringe and injected in a Shimazhu GC-2014 gas chromatograph, with Ar as the carrier gas. Hydrogen was detected with a thermal conductivity detector, referring to the standard hydrogen gases with known concentrations. Increased pressure from the evolved hydrogen is neglected in the calculations.

### The apparent quantum yields measurement

The apparent quantum yields (AQYs) were obtained according to the following equation;

$$AQY\,(\%) = \frac{2 \times \text{Number of evolved H}_2\text{molecules}}{\text{Number of incident photons}} \times 100\% \qquad (1)$$

$$= \frac{2 \times M \times N_A}{S \times P \times t \times \lambda/(h \times c)} \times 100\% \qquad (2)$$

Where, M is the amount of hydrogen evolution; N$_A$ is Avogadro constant; h is Planck constant; c is light velocity; S is the irradiation area(5cm$^2$ in our experiment); P is the incident light intensity (at 420, 500, 550, 600 and 700 nm are 20, 40, 50, 70, and 100 W m$^{-2}$, respectively, in our experiment); t is the time of light irradiation; λ is the wavelength of monochromatic light, (420, 500, 550, 600 and 700 nm in our experiment). For the AQYs measurement, the mixed solutions were consisted of Pdot photocatalyst (5 mg/10 mL water), ascorbic acid (AA, 0.1 M), and 3 wt% H$_2$PtCl$_6$, it was illuminated with light source of a Xenon lamp (AM1.5).

### Characterizations.

$^{1}$H and $^{13}$C NMR spectra were measured using a Bruker Avance 500 MHz NMR spectrometer. TGA of the polymers was performed under N$_2$ using a TA Q600 instrument over the temperature range 40–550 °C (heating rate: 20 °C min$^{-1}$). UV–Vis absorption spectra of the polymers were recorded using Hitachi U-3300 and Dynamica HALO DB-20S spectrophotometers. Fluorescence spectra of the polymers were recorded using a Hitachi F-7000 spectrophotometer at room temperature. The energy levels of the HOMOs were measured using a photoelectron spectrometer (model AC-2). The optical bandgap (Eg) is derived by Tauc Plots of (αhv)$^{2}$ versus (hv) from the UV-Vis spectra. The energy levels of the LUMOs were calculated by subtracting the Eg from the HOMO energy levels. X-ray photoelectron

spectroscopy (XPS) spectra were collected using a ULVAC-PHI PHI 5000 Versaprobe II chemical analysis electron spectrometer (ESCA). The polymers thin films were prepared from drop-casting on silicon wafer substrates for XPS measurement.

### Transient absorption spectroscopy measurement.

Time-resolved experiments were performed using laser-based spectroscopy, with a laser power of less than one photon absorption per particle. Samples for transient absorption experiments were prepared as polymer nanoparticles solution (1 mg /10 mL) and kept in the dark between measurements. A Coherent Legend Ti: Sapphire amplifier (800 nm, 100 fs pulse length, 3 kHz repetition rate) was used. The output was split to pump and probe beams. Excitation pulses at specific wavelengths were acquired using an optical parametric amplifier (Topas C, Light Conversion). The probe pulses (a broad supercontinuum spectrum) were generated from the 800-nm pulses in a CaF$_2$ crystal and split by a beam splitter into a probe pulse and a reference pulse. The probe pulse and the reference pulse were dispersed in a spectrograph and detected by a diode array. The instrumental response time was approximately 100 fs. The kinetic traces were fitted with a sum of convoluted exponentials:

$$Y(t) = \text{ext}\left[-\frac{(t - t_0)^2}{\tau_p}\right] * \sum_i A_i \exp\left(-\frac{t - t_0}{\tau_i}\right) \qquad (3)$$

where $\tau_p = \frac{IRF}{2\ln 2}$ and IRF is the width of the instrument response function (full width at half-maximum), t$_0$ is the time zero, A$_i$ and τ$_i$ are amplitude and decay times, respectively, and * is the convolution operator.

To avoid non-linear exciton recombination, we used low photon flux to excite one exciton/Pdot similar to different quantum dots-based systems. We used two different wavelengths to excite the studied systems 650 nm to pump PITIC-ThF and PITIC-Th and 550 nm to pump PITIC-Ph (Supplementary Fig. 44).

### Cryo-TEM and Cryo-ED.

The nano-polymer morphologies were examined by a FEI Tecnai G2 F20 bioTWIN Transmission Electron Microscope at 200 keV. Four μL of the sample containing -1 mg/ml nanoparticles were pipetted onto a glow-discharged (15 s on the carbon side) 200-mesh copper grids (HC200-Cu, PELCO), which were blotted in a chamber at 100% humidity at 4 °C for 3 s and plunge-frozen into liquid ethane cooled by liquid nitrogen using a Vitrobot (FEI, Hillsboro, OR). The grid was stored in liquid nitrogen until mounted for imaging. Cryo-transmission electron microscopy images were recorded at a defocus of -1-1.2 μm under low-dose exposures (2800 e/nm$^2$.s) with a 4kx4k charge-coupled device camera (Glatan, Pleasanton, CA) at a magnification of 80,000X. Images and electron diffraction patterns were recorded in the low-dose mode to minimize electron beam radiation damage to the very radiation-sensitive samples.

### Electrochemical Impedance Spectroscopy.

The Zahner Zennium E workstation, which featured a three-electrode cell consisting of a Pt wire counter electrode, an Ag/AgCl reference electrode (3 M NaCl), and a fluorine-doped tin oxide (FTO) glass working electrode, was utilized to perform Electrochemical Impedance Spectroscopy (EIS) measurements. An active area of 1 cm$^2$ FTO glass was drop-casted with 0.5 mL of a Pdot solution (1 mg/mL) at a temperature of 70–80 °C. EIS spectra were collected at a voltage of 1.5 V$_{Ag/AgCl}$ in 0.5 M Na$_2$SO$_4$ electrolyte solution with an amplitude of 20 mV and a frequency range of 1 Hz to 100 kHz, while the sample was illuminated with a white light-emitting diode (LED) PAR38 lamp (20 W, 6500 K, Zenaro Lighting; λ > 420 nm).

## Grazing-incidence wide-angle X-ray scattering (GIWAXS) measurement

GIWAXS in a grazing-incidence geometry at angle of 0.12o was performed to characterize the internal chain segmental packing of PBTIC-TH and PBTIC-THF thin films with a X-ray beam of photon energy, 15 keV, and wavelength, 1.027 Å, at the Beamline TPS13A1 of the National Synchrotron Radiation Research Center (NSRRC), Taiwan.

**Computational details.** We constructed an ADA-L-ADA dimer model to simulate the properties of each polymer. All geometry optimizations were performed using DFT at the $\omega$B97X-D/6-31 G(d) level of theory. The excited-state computations were performed using TD-DFT with Tamm-Dancoff Approximation at $\omega$B97X-D/6-31 G(d) level of theory. The dihedral scanning was performed using a constrained-optimization scheme along the linker-acceptor dihedral defined in Fig. 4. The $\lambda_{S_1 \to S_0}$ is computed using the following equation:

$$\lambda_{S_1 \to S_0} = E_{S_0/S_1} - E_{S_0/S_0} \tag{4}$$

where $E_{S_0/S_1}$ and $E_{S_0/S_0}$ are the $S_0$ energy computed at the geometry of $S_1$ minimum and $S_0$ minimum, respectively. All aforementioned computations were computed using GAUSSIAN16. Finally, the electronic coupling between the two neighboring acceptors connected through a linker (AD**A**-L-**A**DA) was computed using CI-CDFT at the $\omega$B97X-D/6-31 G(d) level of theory implemented in QChem v5.4.1.

## Data availability

The data that support the findings within this paper are available within the article and the Supplementary Information file, or available from the corresponding author upon request. Source data are provided with this paper.

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

## Acknowledgements

The authors gratefully acknowledge the financial support of National Science and Technology Council of Taiwan (NSTC 112-2223-E-007-006-MY3, 112-2622-E-007-032, 113-2923-E-007-007-MY3, 111-2628-E-007- 009- and 110-2123-M-002-006), Science Vanguard Project of National Science and Technology Council (NSTC) [110-2123-M-002-006 and NSTC 111-2123-M-002-009]; the iMATE program of Academia Sinica [AS-iMATE-110-34]; and the Center of Atomic Initiative for New Materials (AI-Mat), National Taiwan University, from the Featured Areas Research Center Program within the framework of the Higher Education Sprout Project by the Ministry of Education (MOE) of Taiwan [110 L9008 and 111 L9008]. The authors appreciate the Precision Instrument Support Center of National Tsing Hua University in providing the analysis and measurement facilities. The authors appreciate for Dr. Yuan-Chih Chang for cryo-EM experiments which performed at the Academia Sinica Cryo-EM Facility (ASCEM). K.-H.L. and H.-H.C. thank National Center for High-performance Computing (NCHC) for providing computational and storage resources.

## Author contributions

M.H.E. conceived the project, designed, planned, and performed the experiments, prepared the polymer photocatalysts, wrote the manuscript, and discussed the results with all authors. M.A., B.-H.C., C.-H.L. and S.-D.Y. carried the transient absorption spectroscopy experiments. A.Z.A., A.E.A.A. and C.-H.H. performed the electrochemical experiments. I.M.A.M., M.G.M., S.-W.K. and A.S. assisted in the characterizations. K.H.L. performed the calculations. W.-S.C., S.-J.L. and C.-Y.C. performed the SAXS/WAXS experiments. L.-C.C. and K.-H.C. assisted by providing resources and reviewing the writing of the paper. H.-H.C. supervised the work on polymer synthesis and photocatalysis. All the authors participated in the interpretation and discussion of the results.

## Competing interests

The authors declare no competing interests.

## Additional information

[1]Department of Chemistry, Faculty of Science, Al-Azhar University, Nasr City 11884 Cairo, Egypt. [2]Institute of Atomic and Molecular Sciences, Academia Sinica, Taipei 10617, Taiwan. [3]Center for Condensed Matter Sciences, National Taiwan University, Taipei 10617, Taiwan. [4]Department of Chemical Engineering, National Tsing Hua University, Hsinchu 300044, Taiwan. [5]Department of Chemistry, United Arab Emirates University, Al Ain P.O. Box 15551, United Arab Emirates. [6]Department of Chemistry, Qena Faculty of Science, South Valley University, 83523 Qena, Egypt. [7]Chemical Physics and NanoLund, Lund University, 22100 Lund, Sweden. [8]Chemistry Department, Faculty of Science, Al-Azhar University, Assiut 71542, Egypt. [9]Sustainable Chemical Science and Technology, Taiwan International Graduate Program, Taipei, Taiwan. [10]Department of Applied Chemistry, National Yang Ming Chiao Tung University, Hsinchu 30010, Taiwan. [11]Institute of Chemistry, Academia Sinica128 Sec 2 Academia Rd., Nankang, Taipei 11529, Taiwan. [12]Institute of Photonics Technologies, National Tsing Hua University, Hsinchu 300044, Taiwan. [13]Center of Atomic Initiative for New Materials, National Taiwan University, Taipei 10617, Taiwan. [14]Department of Chemical Engineering, National Chung Hsing University, Taichung 40227, Taiwan. [15]Department of Materials and Optoelectronic Science, Center for Functional Polymers and Supramolecular Materials, National Sun Yat-Sen University, Kaohsiung 804, Kaohsiung, Taiwan. [16]Department of Physics, National Taiwan University, Taipei 10617, Taiwan. ✉e-mail: hhchou@mx.nthu.edu.tw

