## [Peer review file · Nature Communications]

REVIEWER COMMENTS

Reviewer #1 (Remarks to the Author):

This study makes an important contribution to the area and as such I have read it with interest. I wish though that the trade-off in terms of loss of driving force would have been discussed in the manuscript. The acceptors are fantastic in OPV cells but in this application the shift of the positions causes significant issues going forward. I have also a list of other comments, some technical but also many that relate to presentation and lack of details that should have been included.

Overall, I believe that it has the potential to be accepted for publication after suitable revision.

1) The introduction should also state the absolute limits in terms of energetics to drive overall water splitting, and half-reactions.

2) The water oxidation potential and oxidation potentials of ascorbic acid should also be added to figure 2c.

3) Work on overall water splitting with organic materials/conjugated polymers should have been included as part of the introduction. It is important to acknowledge that there are huge challenges in making these systems work beyond scavengers.

4) A forward looking discussion is missing as the HOMO levels sit significantly above the water oxidation potential. How will this challenge be addressed? I don't really look for a total solution as such here but it is not clear to me how these systems would perform overall water splitting and a clear trade-off exists in using NIR light here as the materials design shifts the potentials up. Acknowledging the challenge and ways to overcome these would improve the manuscript significantly.

5) I am unclear about the meaning of the following statement "This result is strong evidence to prove that the high photocatalytic activity of our materials is truly beneficial for the development of hydrogen production under visible light."

6) A direct comparison to the state-of-the-art in terms of AQYs/EQEs is very important. These rates are not comparable and a comparison to other material classes, in particular other polymer classes but also to inorganic materials should be partially at least presented in the main-text. Currently rates are directly compared which is simply not possible in this area.

7) Why were different light sources used within the study? I would have liked to have seen at least some data for both visible and NIR measurements from one light source.

8) The TA data focuses very much on excited state lifetime. Durant has shown that the scavenger is potentially not innocent and can assist exciton separation. I would like to see at least some data for this given that the materials are different from D/A systems by McCulloch that have separate domains.

9) How stable are these samples? They appear to have reduced activity within 4 hours; Do they remain active after 24 hours?

- 10) Data to show stability of the materials in terms of structure and particles should also be presented.
- 11) Absolute hydrogen evolution rates should be presented. I am also confused by the number presented throughout the tables, text, conclusions and abstract. Some of these do not seem to match up. I am not questioning that the materials are very active, but I am unclear about their value.
- 12) Figure 2b) is cut-off just below 900 nm which means that the full absorption onset is not visible.
- 13) One significant issue that I have with this work is that figure S26 is shown to vaguely indicate that the reaction is metal free. Palladium will be present in the materials (and should have been quantified) which will act as the proton reduction site. Without significant additional (experimental) evidence of intermediates I cannot see this adding much to the manuscript.
- 14) How much THF remains in the dispersion used for the photocatalytic experiments? Does this add to the activity?
- 15) The models used for the HOMO/LUMO predictions are potentially too small. The authors should show at least for one of the materials that no changes occur upon adding additional monomer units.
- 16) How are the PL spectra in figure S27 comparable given that nanoparticles are present that scatter?
- 17) Details relating to the electrochemical impedance spectroscopy measurements are missing (how were these experiments set up? What circuit configuration was used?). Where the nanomaterials studied? If films were used how do the authors ensure that they are representative of the nanomaterials? How was the contact between the material and electrode characterized given that this is an interface which does not exist in photocatalysis? The discussion is also not very detailed and should provide values for resistance, capacitance, Warburg impedance, and the charge carrier transfer resistance rather than making handwavy statements.
- 18) The ESI is lacking many details that would allow other researcher to reproduce the results.
- a) The synthesis section only provides proton NMR. For any synthetic work this is not acceptable and as a bare minimum CHN analysis and mass spec data has to be provided to show that the intermediates and monomers are pure and of the suggested composition.
- b) For an ESI the use of abbreviations is excessive and full names should be used in the procedures to make it clearer to the reader what was used without constantly cross-referencing.
- c) The synthetic procedures for the polymerisations also lack detail. What amounts and stoichiometry was used? For example, it states Monomer 'Br-ITIC-Br or Br-BTIC-Br (152 mg 0.1 mmol)' which indicates only the amounts and stoichiometry of the second monomer.
- d) The procedure states that a 'sealed tube' was used. Is this the case? Given the temperature above the boiling point of both water and toluene this should actually be noted to be of risk to others in the text. I have seen many student producing bombs which caused terrible accidents, some with injuries that could have been prevented.
- e) For the polymerisations no yields are given.
- 19) Why do the materials do not fully decompose at temperatures exceeding 500 deg C?

Reviewer #2 (Remarks to the Author):

This work reports polymers with absorption in NIR region for photocatalytic hydrogen production and aims to study the charge separation and recombination processes. The synthesized polymers are interesting and the photocatalytic performance is good. However, there are much data which can not well support the discussion and conclusion.

1. Impedance results are used to explain that PS-PEG-COOH/PITIC-ThF Pdots has large resistance than PITIC-ThF Pdot because the surfactant hinders the interaction between polymers. This is not correct. The impedance mainly measured the resistance from interface charge transfer. The Pdots with surfactant of course have long hydrophilic arms which will weaken the interaction with the electrode. But it does not mean the inner interaction between polymers is weak.

2. Figure 3 g and h provide misleading information, one can not simply compare EQE and hydrogen generation rate per gram catalyst. Unless all systems used same amount catalyst in a certain volume.

3. From the title, this work should focus on charge recombination study, however, TAS study is superficial and does not fully support the conclusion. The conclusion of charge recombination with the order of ThF > Th > Ph is not well supported from Figure 5, ThF and Th actually show very similar lifetime. Also, in photocatalysis, the system has sacrificial donor and Pt, the charge recombination kinetics should be different from the pure Pdots system. One should not simply compare the charge recombination in a pure Pdots system and apply the conclusion to a photocatalytic system.

4. It is not good to normalize the amplitude of TAS data and then only compare the kinetics. Actually, Ph showed much large amplitude that means the system generates more excitons than other polymers (if the absorbance is the same).

5. Prepare polymer particles without surfactant is well-known. The author should not claim it as "our method"

6. Figure 6b, the comparison is not informative. It is good to compare the same Pdots with or without Pt.

7. Why not to study the reductive quenching from ascorbic acid to Pdots? If the author want to provide more information on charge recombination as stated in the title.

8. Why did the author use different Pdots concentrations for visible and NIR experiments? No explanations found. The Pdots concentration needs to be provided in different experiments.

9. The author need to well proof-read the manuscript, there are many grammar errors and typos.

Response to the reviewers' comments

We express our gratitude to the reviewers for their thorough review of our manuscript. We have carefully considered their comments, which we deem to be highly relevant and significant, and have made appropriate changes to the paper in response.

Our point-by-point response to the reviewers' comments (highlighted in **blue**) and their comments and remarks (in **black**) is presented below, along with a detailed explanation of the corresponding revisions and additions that we have incorporated to adequately address them in the revised manuscript and Supplementary Information.

Responses to Reviewer #1

Comments to the Author

Comments:

This study makes an important contribution to the area and as such I have read it with interest. The acceptors are fantastic in OPV cells but in this application the shift of the positions causes significant issues going forward. I have also a list of other comments, some technical but also many that relate to presentation and lack of details that should have been included.

Overall, I believe that it has the potential to be accepted for publication after suitable revision.

Response: We express our appreciation to Reviewer #1 for their positive feedback, constructive suggestions, and overall favorable evaluation of our work. Our responses to their specific queries are detailed below.

Comment 1. The introduction should also state the absolute limits in terms of energetics to drive overall water splitting, and half-reactions.

Response: Thank you very much for the reviewer's note. Per the reviewer's suggestion we modified the introduction as the following on page 4 of the revised manuscript.

“In an ideal scenario, a photocatalyst must be a semiconductor material that possesses a band gap energy equal to or greater than the electrochemical potential (1.23 eV) necessary for propelling water-splitting reactions. However, for practical purposes, the band gap of a photocatalyst must be in the range of 1.8 to 2.0 eV.⁶ Nevertheless, this requirement alone is insufficient for enabling overall water-splitting reactions since different combinations of conduction band (CB) and valence band (VB) can result in the same band gap energy. To fulfill the criteria, CB must have a more negative potential than that of the proton reduction reaction, while VB must have a more positive potential than that of the water

oxidation reaction.^{7, 8, 9, 10} The demanding specifications of the photocatalyst material, therefore, pose significant limitations and challenges in the selection of suitable materials for overall water-splitting reactions.^{11, 12} On the other hand, numerous studies have reported the effective utilization of a sacrificial hole scavenger to enhance the photocatalytic production of hydrogen from water.”

References;

6. Rahman M, Tian H, Edvinsson T. Revisiting the Limiting Factors for Overall Water-Splitting on Organic Photocatalysts. *Angew Chem Int Ed Engl* 59, 16278-16293 (2020).
7. Sprick RS, et al. Water Oxidation with Cobalt-Loaded Linear Conjugated Polymer Photocatalysts. *Angew Chem Int Ed Engl* 59, 18695-18700 (2020).
8. Bai Y, et al. Photocatalytic overall water splitting under visible light enabled by a particulate conjugated polymer loaded with palladium and iridium. 134, e202201299 (2022).
9. Chen HM, et al. Covalent triazine-based frameworks with cobalt-loading for visible light-driven photocatalytic water oxidation. *Catalysis Science & Technology* 12, 5442-5452 (2022).
10. Bai Y, et al. Photocatalyst Z-scheme system composed of a linear conjugated polymer and BiVO₄ for overall water splitting under visible light. *Journal of Materials Chemistry A* 8, 16283-16290 (2020).
11. Bai Y, Hippalgaonkar K, Sprick RS. Organic materials as photocatalysts for water splitting. *Journal of Materials Chemistry A* 9, 16222-16232 (2021).
12. Saunders B, Wilbraham L, Prentice AW, Sprick RS, Zwiijnenburg MA. The potential scarcity, or not, of polymeric overall water splitting photocatalysts. *Sustainable Energy & Fuels* 6, 2233-2242 (2022).

Comment 2. The water oxidation potential and oxidation potentials of ascorbic acid should also be added to figure 2c.

Response: We appreciate the reviewer’s valuable feedback. As per the suggestion of the reviewer, we have incorporated the water oxidation potential and oxidation potential of ascorbic acid into figure 2c. As results, the modified Figure 2c was added in the revised manuscript on page 10:

Figure 2c: Energy level diagrams of the all-polymers were measured using a photoelectron spectrometer. The dashed lines correspond to the proton reduction potential (H^+/H_2), water oxidation potential (O_2/H_2O) and the calculated potential of the two-hole (A/H_2A) and one-hole (HA/H_2A) oxidation of ascorbic acid at pH 2.6 (the experimentally measured pH of 0.1 M ascorbic acid solution).⁴ All energy levels and electrochemical potentials are expressed relative to vacuum (using -4.44 V versus vacuum as equivalent to 0 V versus SHE).¹⁶

4. Wang X, et al. Sulfone-containing covalent organic frameworks for photocatalytic hydrogen evolution from water. *Nature Chemistry* 10, 1180-1189 (2018).

16. Kosco J, et al. Generation of long-lived charges in organic semiconductor heterojunction nanoparticles for efficient photocatalytic hydrogen evolution. *Nature Energy* 7, 340-351 (2022).

Comment 3. Work on overall water splitting with organic materials/conjugated polymers should have been included as part of the introduction. It is important to acknowledge that there are huge challenges in making these systems work beyond scavengers.

Response: We appreciate the reviewer's note. Per the reviewer's suggestion we modified the introduction as the following on page 4 of the revised manuscript,

"In an ideal scenario, a photocatalyst must be a semiconductor material that possesses a band gap energy equal to or greater than the electrochemical potential (1.23 eV) necessary for propelling water-splitting reactions. However, for practical purposes, the band gap of a photocatalyst must be in the range of 1.8 to 2.0 eV.⁶ Nevertheless, this requirement alone is insufficient for enabling overall water-splitting reactions since different combinations of conduction band (CB) and valence band (VB) can result in the same band gap energy. To fulfill the criteria, CB must have a more negative potential than that of the proton reduction reaction, while VB must have a more positive potential than that of the water

oxidation reaction.^{7, 8, 9, 10} The demanding specifications of the photocatalyst material, therefore, pose significant limitations and challenges in the selection of suitable materials for overall water-splitting reactions.^{11, 12} On the other hand, numerous studies have reported the effective utilization of a sacrificial hole scavenger to enhance the photocatalytic production of hydrogen from water.”

References;

6. Rahman M, Tian H, Edvinsson T. Revisiting the Limiting Factors for Overall Water-Splitting on Organic Photocatalysts. *Angew Chem Int Ed Engl* 59, 16278-16293 (2020).
7. Sprick RS, et al. Water Oxidation with Cobalt-Loaded Linear Conjugated Polymer Photocatalysts. *Angew Chem Int Ed Engl* 59, 18695-18700 (2020).
8. Bai Y, et al. Photocatalytic overall water splitting under visible light enabled by a particulate conjugated polymer loaded with palladium and iridium. 134, e202201299 (2022).
9. Chen HM, et al. Covalent triazine-based frameworks with cobalt-loading for visible light-driven photocatalytic water oxidation. *Catalysis Science & Technology* 12, 5442-5452 (2022).
10. Bai Y, et al. Photocatalyst Z-scheme system composed of a linear conjugated polymer and BiVO₄ for overall water splitting under visible light. *Journal of Materials Chemistry A* 8, 16283-16290 (2020).
11. Bai Y, Hippalgaonkar K, Sprick RS. Organic materials as photocatalysts for water splitting. *Journal of Materials Chemistry A* 9, 16222-16232 (2021).
12. Saunders B, Wilbraham L, Prentice AW, Sprick RS, Zwiijnenburg MA. The potential scarcity, or not, of polymeric overall water splitting photocatalysts. *Sustainable Energy & Fuels* 6, 2233-2242 (2022).

Comment 4. A forward-looking discussion is missing as the HOMO levels sit significantly above the water oxidation potential. How will this challenge be addressed? I don't really look for a total solution as such here but it is not clear to me how these systems would perform overall water splitting and a clear trade-off exists in using NIR light here as the materials design shifts the potentials up. Acknowledging the challenge and ways to overcome these would improve the manuscript significantly.

Response: We thank reviewer for pointing this out. The reviewer's observation regarding the crucial role of the overall water splitting reaction is one with which we wholeheartedly agree. Moreover, we acknowledge that the development of organic materials to successfully achieve this reaction remains a formidable challenge. Our designated materials have HOMO levels situated below the water oxidation potential and LUMO levels above the hydrogen reduction potential in the photocatalytic conditions (pH = 2.6, acidic), as depicted in Figure S29a. This suggests that these materials are likely to succeed in overall water splitting under acidic conditions. While it is still possible to carry out water splitting in acidic solutions, it is generally more challenging due to the higher overpotential, and the protonation of catalysts. Neutral solutions are typically preferred for water splitting reactions due to their more favorable conditions. In a neutral medium, the LUMO level of most of our designated polymers is lower than the proton reduction reaction, and the HOMO level is significantly lower than the water oxidation reaction (Figure S29b), making most of these materials only suitable for water oxidation. To surmount the challenges of achieving overall water splitting with our designated materials, one possible solution by following the way of Prof. Sprick and Prof. Cooper groups by hybridizing our polymers with other

active materials for proton reduction to produce a Z-scheme photocatalytic system could be able to achieve the overall water splitting and NIR activity simultaneously.

Per the reviewer's suggestion we add the following sentences to the manuscript for acknowledging the challenge and ways to overcome (page 13-14 of the revised manuscript).

“While our designated materials demonstrate promising HER and AQY% efficiency with a sacrificial (AA) half reaction under both visible and NIR light, they face significant challenges in achieving overall water splitting. This is because the presented polymers have HOMO levels that are below the water oxidation potential and LUMO levels that are above the hydrogen reduction potential under photocatalytic conditions (pH = 2.6, acidic), as depicted in **Figure S29a**. Thus, these materials are more likely to succeed in overall water splitting under acidic conditions, which is generally more challenging due to higher overpotential, and protonation of catalysts. In contrast, neutral solutions are typically preferred for water splitting reactions because of their more favorable conditions. However, in a neutral medium, most of our designated polymers have a LUMO level that is lower than the proton reduction reaction, and a HOMO level that is significantly lower than the water oxidation reaction (**Figure S29b**), making them only suitable for water oxidation. To overcome the challenges of achieving overall water splitting with our designated materials, one possible solution is to hybridize them with other active materials for H₂ reduction, as Prof. Sprik and Prof. Cooper's groups have done, to produce a Z-scheme photocatalytic system that could achieve overall water splitting and NIR activity simultaneously.”

Figure S29: Energy level diagrams of the all-polymers were measured using a photoelectron spectrometer. The dashed lines correspond to the proton reduction potential (H^+/H_2), water oxidation potential (O_2/H_2O) (a) in acidic medium (pH = 2.6), (b) in neutral medium (pH = 7).⁴ All energy levels and electrochemical potentials are expressed relative to vacuum (using -4.44 V versus vacuum as equivalent to 0 V versus SHE).¹⁶

4. Wang X, et al. Sulfone-containing covalent organic frameworks for photocatalytic hydrogen evolution from water. *Nature Chemistry* 10, 1180-1189 (2018).

16. Kosco J, et al. Generation of long-lived charges in organic semiconductor heterojunction

nanoparticles for efficient photocatalytic hydrogen evolution. *Nature Energy* 7, 340-351 (2022).

Comment 5. I am unclear about the meaning of the following statement “This result is strong evidence to prove that the high photocatalytic activity of our materials is truly beneficial for the development of hydrogen production under visible light.”

Response: We thank the reviewer for bringing this to our attention. We apologize for any confusion caused by this statement. What we meant to convey is that the high photocatalytic activity of our materials, as demonstrated in our experiments, suggests that they have great potential for use in the development of efficient hydrogen production systems that utilize visible light. However, we understand that this statement may have been unclear, then we clarify it in the manuscript as the following.

“Our photocatalyst demonstrates a significant HER value, suggesting its potential as an effective catalyst for hydrogen production when exposed to visible and NIR light.”

Comment 6. A direct comparison to the state-of-the in terms of AQYs/EQEs is very important. These rates are not comparable and a comparison to other material classes, in particular other polymer classes but also to inorganic materials should be partially at least presented in the main-text. Currently rates are directly compared which is simply not possible in this area.

Response: Thanks for the valuable feedback of the reviewer on the direct comparison of AQYs/EQEs. We acknowledge that experimental factors can influence the measurement of HER, making it challenging to directly compare results across different studies. To provide transparency and clarity, we will include the different conditions used in other articles. Moreover, as recommended, we summarized the AQYs/EQEs of different material categories to offer a more comprehensive assessment of the new materials' performance. Table S4 presents a summary of various polymer photocatalysts, including their respective photocatalytic reaction conditions. In addition, we summarize other organic and inorganic photocatalysts in terms of their photocatalytic activity under both visible and near-infrared (NIR) light (Table S5).

Table S4. Comparative studies of our developed polymer photocatalyst versus other polymer photocatalysts in terms of HER, AQY% with the reaction conditions and light source.

Polymer ^a	Conditions	Light Source	HER (mmol h ⁻¹ g ⁻¹)	AQY % at wavelengths (nm)	References
PFBT- Pdots	0.075 mg in 3 mL of 0.2M ascorbic acid.	A LED lamp (λ > 420 nm)	8.3	0.5 at 420 nm	Angew. Chem. Int. Ed. 2016, 55, 12306–12310
PFODTBT Pdot	0.075 mg in 3 mL of 0.2M ascorbic acid.	A LED lamp (λ > 420 nm)	50.0	0.9, 0.3, 0.6, and 0.3 at 420, 500, 550, and 600 nm	Energy Environ. Sci., 2017,10, 1372-1376

PFTFQ-PtPy15	1 mg in 10 mL Water/ 20 vol% TEA	A LED lamp ($\lambda > 420$ nm)	12.7	0.4 at 500 nm	ACS Catal. 2018, 8, 7766–7772
F8T2 Pdots/g-C3N4	20 mg in (90 mL Water + 10 mL TEOA)	300 W Xenon lamp ($\lambda > 400$ nm)	0.93	5.7, 2.8, and 0.8 at 420, 500, and 550 nm	J. Mater. Chem. A, 2019, 7, 303-311
HE-CP10-Dots	20 mg + 55 mL water + ascorbic acid (1.76 g)	300 W Xenon lamp ($\lambda > 420$ nm)	0.84	0.9 at 500 nm	Macromolecules 2019, 52, 11, 4376–4384
F8DTBT Pdots/CNN S	20 mg in (90 mL Water + 10 mL TEOA)	300 W Xenon lamp ($\lambda > 400$ nm)	0.181	3.4, 0.4, 0.2, and 0.5 at 420, 500, 550, and 600 nm	Appl. Catal. B: Environ. 2019, 259, 118067
PFN-Br	2.5 mg in (5 mL TEOA + 45 mL water)	300-W Xe lamp ($\lambda > 300$ nm)	0.68	0.12, 0.40, 0.44 and 0.19 at 550, 600, 650 and 700 nm	Nano Energy, 2019, 60, 775–783
PBDBTBT-7EO (3.0 wt% Pt)	2.5 mg in 50 mL of AA solution (0.2 M)	300-W Xe lamp ($\lambda > 300$ nm)	15.9	0.13, 0.14, 0.25, and 0.30 at 420, 500, 550, and 600 nm	iScience, 2019, 13, 33–42.
PFTBTA-PtPy	1 mg in 10 mL Water/ 20 vol% TEA	A LED lamp ($\lambda > 420$ nm)	7.34	0.5 at 420 nm	Appl. Catal. B: Environ. 2020, 268, 118436
PFNBtBr Pdots/CNN S	20 mg in (90 mL Water + 10 mL TEOA)	300 W Xe lamp ($\lambda > 400$ nm)	1.2	7.71, 2.5, 2.0 at 420, 500, and 550 nm	Appl. Catal. B: Environ. 2020, 270, 118852
PTB7-Th/EH-IDTBR Pdot	2 mg in 20 mL 0.2M AA solution	300-W Xe lamp	28.13	2.0, 2.3, 4.3, 5.6, and 6.2 at 420, 500, 620, 660, and 700 nm	Nat. Mater. 2020, 19, 559–565
PyDTDO-3 (w/o Pt)	1.0 M AA solution / 10 vol% DMF	10 mg in (90 mL of 1M AA + 10 mL DMF)	16.32	3.70, 3.68, 3.93, and 2.30 at 420, 500, 550, and 600 nm	Chem. Sci., 2021,12, 1796-1802.
PS-PEG5-FNP	0.2M AA solution	5 mg in (25 mL ascorbic acid (0.2 M))	37.2	2.5 at 420 nm	Angew. Chem. Int. Ed. 2021, 60, 15590–15597.
PyBS-3	25 mg in (100 mL 0.2M AA solution)	300 W Xenon lamp ($\lambda > 300$ nm)	100.1	29.3 at 420 nm	Adv. Mater. 2021, 33, 2008498.

ZnCoP-F CP	30 mg in (42.5 mL water + 7.5 mL TEOA)	300 W Xe-lamp with a cutoff filter ($\lambda \geq 400$ nm)	2.76	6.92, 5.19, 5.50, 5.78, 3.17, 1.93 at 400, 450, 500, 550, 700, and 760 nm	Adv. Funct. Mater. 2021, 31, 2009819.
D1/D2/ITI C	0.062 mg in (1.5 mL water + 0.5 mL AA 0.8 M)	LED PAR38 lamp ($\lambda > 420$ nm)	60.8	2.2, 4.6, 6.5, 7.1, 6.1, 4.1 at 450, 500, 550, 600, 650, and 700 nm	J. Am. Chem. Soc. 2021, 143, 2875.
gIDTBT:oI DTB R	1 mg in (0.2 M AA (12 mL))	Solar simulator (Asahi Max 303) and an AM1.5g filter	18.5	5.3/1.0/2.9/2.8/0. 9% at 400, 440, 620, 660 and 700 nm	Adv. Mater. 2021, 34, 2105007.
PCPDTBS O	2 mg in (1mL NMP + 9 mL ascorbic acid (1M))	300 W Xenon lamp ($\lambda > 350$ nm)	24.6	0.94, 7.77, 8.72, 4.77, 3.74 at 420, 460, 500, 550, and 600 nm	Appl. Catal. B: Environ. 2021, 298, 120577.
PBDTTS- ISO	6 mg in (3mL NMP + 27 mL ascorbic acid (1M))	300 W Xenon lamp ($\lambda > 350$ nm)	97.12	13.5, 16.7, 18.5, and 9.8 at 420, 500, 550, and 600 nm	J. Mater. Chem. A, 2022,10, 6641- 6648
PM6:PCB M 2:8	1 mg in (0.2 M AA (12 ml))	Solar simulator (Asahi Max 303) and an AM1.5g filter	73.7	8.7, 8.8, 7.7, 6.6, 2.6 at 400, 470, 560, 620 and 700 nm	Nat. Energy. 2022, 7, 340-351
PBTIC- ThF Pdot	0.1 mg in (10 mL 0.1M ascorbic acid)	A LED lamp ($\lambda > 420$ nm)	269.4	3.9, 3.2, 3.1, 3.9, and 4.7 at 420, 500, 550, 600 and 700 nm	This work
PITIC- ThF Pdot	0.1 mg in (10 mL 0.1M ascorbic acid)	A LED lamp ($\lambda > 420$ nm)	339.7	2.9, 2.7, 2.5, 2.8, and 3.1 at 420, 500, 550, 600 and 700 nm	This work

Table S5. Comparative studies of our developed polymer photocatalyst versus other photocatalysts in terms of photocatalytic hydrogen evolution under both visible and NIR light.

Photocatalysts	Conditions	Visible light (>420 nm) (mmol g ⁻¹ h ⁻¹)	NIR light (>780 nm) ($\mu\text{mol g}^{-1}\text{h}^{-1}$)	References
Au/La ₂ Ti ₂ O ₇	1.5 mg in 5 mL (1:4) methanol-H ₂ O solution	0.74	300	Angew. Chem. Int. Ed. 2017, 56, 2064–2068
g-C ₃ N ₄ -Co-K	50 mg in (15 mL TEOA + 85 mL H ₂ O).	0.808	470	J. Colloid Interface Sci. 2020, 561, 719–729

WS2@Cu Hybrids	3 mg + 90 mL H ₂ O + 10 mL Lactic Acid + 1g PEG	64	175 (>750 nm)	Adv. Funct. Mater. 2018, 28, 1804055.
Black Phosphorus / g-C ₃ N ₄	1.5 mg in (1 ml methanol + 4 mL H ₂ O)	0.427	101	J. Am. Chem. Soc. 2017, 139, 13234–13242
Au / La ₂ Ti ₂ O ₇	10 mg in (2 mL methanol + 8 mL H ₂ O)	0.34	180	ACS Catal. 2018, 8, 122–131.
g-C ₃ N ₄ / Chlorin e ₆	10 mg in (4 mL TEOA + 16 mL H ₂ O)	1.275	312.6	Appl. Catal. B: Environ. 2020, 260, 118137.
CuNi / rGO composite	5 mg in (60 mL H ₂ O + 10 mL lactic acid)	1.787	86	J. Mater. Chem. A, 2017, 5, 22772–22781.
Black Phosphorus / TiO ₂	2 mg in (1 mL methanol + 4 mL H ₂ O)	0.941	200	ACS Catal. 2019, 9, 3618–3626.
H _{0.53} WO ₃ / CdS – Au	30 mg in (10 mL lactic acid + 90 mL H ₂ O)	10	158	J. Mater. Chem. A, 2019, 7, 1076–1082.
C/K-doped RPCN	20 mg in (3 mL TEOA + 27 mL)	1.4	140	Adv. Mater. 2021, 33, 2101455.
PBTIC-ThF Pdot	0.1 mg in (10 mL 0.1M ascorbic acid)	269.4	---	This work
PBTIC-ThF Pdot	5 mg in (10 mL 0.1M ascorbic acid)	35.6	708	This work
PITIC-ThF Pdot	0.1 mg in (10 mL 0.1M ascorbic acid)	339.7	---	This work
PITIC-ThF Pdot	5 mg in (10 mL 0.1M ascorbic acid)	55.8	4045	This work

Comment 7. Why were different light sources used within the study? I would have liked to have seen at least some data for both visible and NIR measurements from one light source.

Response: We would like to express our gratitude to the reviewer for their comment. Following their suggestion, we performed two separate tests on the HER of our best material (PITIC-ThF Pdot) using visible and NIR light, respectively, while utilizing the same Xenon lamp as the light source. For both experiments, we dispersed 5 mg of PITIC-ThF Pdot in 10 mL of 0.1M AA solution with 3% Pt as the cocatalyst. A light cut filter at $\lambda > 380$ nm and a light intensity of 1000W/m² were used in the visible-light experiment, while in the NIR experiment, a light cut filter at $\lambda > 780$ nm and a light intensity of 3000W/m² were employed. In Figure S26, PITIC-ThF Pdot achieved a HER of 171.3 $\mu\text{mol/h}$ under visible light and 20.2 $\mu\text{mol/h}$ under NIR light using the Xenon lamp as the light source.

Figure S26: Time course of the produced H_2 for the PITIC-ThF under NIR light and Visible light using the same light source (Xe lamp). We performed two separate tests on the HER of our best material (PITIC-ThF Pdot) using visible and NIR light, respectively, while utilizing the same Xenon lamp as the light source. For both experiments, we dispersed 5.0 mg of PITIC-ThF Pdot in 10 mL of 0.1M AA solution with 3% Pt as the cocatalyst. A light cut filter at $\lambda > 380$ nm and a light intensity of 1000 W/m^2 were used in the visible light experiment, while in the NIR experiment, a light cut filter at $\lambda > 780$ nm and a light intensity of 3000 W/m^2 were employed. In Figure S26, PITIC-ThF Pdot achieved a HER of $171.3 \mu\text{mol/h}$ under visible light and $20.2 \mu\text{mol/h}$ under NIR light using the Xenon lamp as the light source.

Comment 8. The TA data focuses very much excited state lifetime. Durrant has shown that the scavenger is potentially not innocent and can assist exciton separation. I would like to see at least some data for this given that the materials are different from D/A systems by McCulloch that have separate domains.

Response: We thank the reviewer for this valuable comment. According to the reviewer's suggestion we study the effect of sacrificial and Pt on our photocatalyst using the TA spectroscopy and the data presented in the following figures. The following discussion and figures are incorporated into the revised manuscript and SI as the following.

“Durrant & McCulloch have reported that the kinetics of charge recombination in the pure Pdot system differ from those in the full photocatalytic system, where the presence of a scavenger and Pt cocatalyst has a significant impact.⁸ To gain a better understanding of the effect of Pt and scavenger on the photocatalytic activity, transient absorption spectroscopy (TAS) was employed. The TAS spectra of neat PITIC-ThF Pdot in the wavelength region (600-980 nm) were recorded after excitation at 650 nm, and the results were analyzed (**Figure 6a**). The TAS spectra of PITIC-ThF Pdot show a broad ground state bleaching (GSB) between 600 nm and 815 nm, with a maximum peak at 750 nm. Additionally, a photoinduced absorption (PIA) was observed starting from 815 nm and reaching a maximum in the NIR region at 959 nm. This PIA is attributed to the singlet exciton absorption of PITIC-ThF Pdot. In **Figure 6b**, the TAS spectra reveals a larger amplitude with the addition of 3% Pt to the Pdot solution, which is consistent with suppressed bimolecular recombination due to electron transfer to Pt. On the other hand,

the addition of AA for Pdot samples with Pt strongly reduced the amplitude in **Figures 6c**, indicating efficient hole extraction in the photocatalytic system.⁸ The exciton decay dynamics of neat PITIC-ThF Pdot and the photocatalytic system containing Pt or Pt + AA were compared in **Figure 6e**. It was found that the addition of Pt to PITIC-ThF Pdot resulted in a longer-lived decay transient compared to the neat PITIC-ThF Pdot, which is consistent with slower bimolecular recombination kinetics with Pt. The further addition of AA resulted in an accelerated decay of the PITIC-ThF Pdot absorption, consistent with hole transfer to AA.^{9, 10} This acceleration was increased even more with AA alone, indicating that the presence of AA scavenger in this photocatalytic system is crucial for achieving high photocatalytic hydrogen evolution rate (**Figure S38**). The same behavior was noticed with PITIC-Th Pdot in the presence of Pt or AA + Pt as presented in **figure S39**. These results provide unequivocal evidence that the presence of AA retards the recombination of photo-generated charge carriers, leading to a nearly three-fold increase in recombination time in the presence of AA and PITIC-ThF Pdot. These results provide insight into the mechanisms of the photocatalytic system and could be useful for the design of more efficient photocatalysts.

Figure 6: Transient absorption spectra for various samples, including pure PITIC-ThF Pdot (a), PITIC-ThF Pdot with 3 wt% Pt (b), and PITIC-ThF Pdot with 0.1M AA and 3 wt% Pt (c), at different time delays after excitation at 650 nm with a power of 0.9 μW . The decay dynamics of the transient absorption were compared for neat PITIC-ThF Pdot, PITIC-ThF Pdot + Pt, and PITIC-ThF Pdot + AA + Pt, when excited at 650 nm and probed at 959 nm, which were assigned to PITIC-ThF Pdot exciton

decay (d).

Figure S38: Transient absorption spectra for PITIC-ThF Pdot with 0.1M AA at different time delays after excitation at 650 nm with a power of $0.9 \mu\text{W}$.

Figure S39: Transient absorption spectra for various samples, including pure PITIC-Th Pdot (a), PITIC-

Th Pdot with 3 wt% Pt (b), and PITIC-Th Pdot with 0.1M AA and 3 wt% Pt (c), at different time delays after excitation at 650 nm with a power of 0.9 μ W. The decay dynamics of the transient absorption were compared for neat PITIC-Th Pdot, PITIC-Th Pdot + Pt, and PITIC-Th Pdot + AA + Pt, when excited at 650 nm and probed at 964 nm, which were assigned to PITIC-Th Pdot exciton decay (d).

The TAS spectra of PITIC-Th Pdot show a broad ground state bleaching (GSB) between 600 nm and 810 nm, with a maximum peak at 740 nm. Additionally, a photoinduced absorption (PIA) was observed starting from 810 nm and reaching a maximum in the NIR region at 964 nm. This PIA is attributed to the singlet exciton absorption of PITIC-ThF Pdot (**Figure S39a**). In **Figure S39b**, the TAS spectra reveals a larger amplitude with the addition of 3% Pt to the Pdot solution, which is consistent with suppressed bimolecular recombination due to electron transfer to Pt. On the other hand, the addition of AA for Pdot samples with Pt strongly reduced the amplitude in **Figures S39c**, indicating efficient hole extraction in the photocatalytic system. The exciton decay dynamics of neat PITIC-ThF Pdot and the photocatalytic system containing Pt or Pt+AA were compared in **Figure S39e**. It was found that the addition of Pt to PITIC-ThF Pdot resulted in a longer-lived decay transient compared to the neat PITIC-ThF Pdot, which is consistent with slower bimolecular recombination kinetics with Pt. The further addition of AA resulted in an accelerated decay of the PITIC-ThF Pdot absorption, consistent with hole transfer to AA.

Comment 9. How stable are these samples? They appear to reduced activity within 4 hours; Do they remain active after 24 hours?

Response: We appreciate the feedback from the reviewer and acted upon his suggestion by performing a long-term stability test for our highest-performing photocatalytic material (PITIC-ThF Pdot), as recommended. The results, as displayed in Figure S27, revealed a hydrogen production of 4115 mmol g^{-1} after 20 hours, which then reached a nearly stable state. We posit that the limited time of H_2 production activity (up to 20 hours) can be attributed to the aggregation of polymer nanoparticles during the photocatalytic experiments. We add the following figure to Supplementary Information for showing the photocatalytic stability.

Figure S27: Stability tests. 24 h H_2 evolution stability tests of PITIC-ThF Pdot. Conditions: ascorbic

acid (AA, 0.1 M), white LED light ($\lambda > 420$ nm, 20 W, and 6500 K), and 3% H_2PtCl_6 . The results, as displayed in the figure, revealed a hydrogen production of $4115 \text{ mmol} \cdot \text{g}^{-1}$ after 20 hours, which then reached a nearly stable state. We posit that the limited time of H_2 production activity (up to 20 hours) can be attributed to the aggregation of polymer nanoparticles during the photocatalytic experiments.

Comment 10. Data to show stability of the materials in terms of structure and particles should also be presented.

Response: Thanks for the reviewer comment. As suggested by the reviewer we test the photocatalytic stability of our best material (PITIC-ThF Pdot) for longer time till 24 h then analyze the photocatalyst after the reaction using FTIR, UV-Vis absorption, and Dynamic Light Scattering (DLS). The result as presented in the following figure which inserted to the Supplementary Information:

Figure S28: (a) Normalized absorption spectra, and (b) fourier transform infrared (FTIR) before and after 24 h of H_2 evolution of PITIC-ThF Pdot. (c) and (d) the dynamic light scattering (DLS) before and after 24 h of H_2 evolution of PITIC-ThF Pdot, respectively.

To investigate the stability of the best material, PITIC-ThF Pdot, the photocatalysts were subjected to a 24-hour extended stability test (as shown in Figure S27). During this test, the photocatalysts were

analyzed using various techniques such as Fourier transform infrared spectroscopy (FTIR), dynamic light scattering (DLS), and UV-Vis absorption analysis, both before and after 24 hours of H₂ evolution. The FTIR analysis showed no change before and after the H₂ evolution reaction (as seen in Figure S28b). However, the UV-Vis absorption analysis showed a slight red shift in the absorption spectrum of PITIC-ThF Pdot after 24 hours of H₂ evolution (as seen in Figure S28a), which suggests that Pdot aggregation may be responsible for the slower reduction in the HER rate. The red shift in the absorption spectrum could be due to the Pdot particles aggregating, as larger particles tend to have a broader absorption spectrum. To confirm this hypothesis, the DLS analysis was conducted. The results showed that the average particle size increased from 30 nm to 1 μm after the H₂ evolution reaction (as seen in Figure S28c, S28d). The increase in particle size indicates that Pdot particles had aggregated, which could be the reason for the slower reduction in the HER rate observed during the stability test.

Comment 11. Absolute hydrogen evolution rates should be presented. I am also confused by the number presented throughout the tables, text, conclusions and abstract. Some of these do not seem to match up. I am not questioning that the materials are very active, but I am unclear about their value.

Response: We extend our appreciation to the reviewer for providing valuable feedback. Based on the reviewer's suggestion, we have included the absolute hydrogen evolution rate (μmol h⁻¹) in the abstract and conclusion. Throughout the main text and table, we have made efforts to present the absolute HER, taking into account the photocatalyst weight effect. To achieve this, we have used two common units for HER presentation, namely mmol g⁻¹h⁻¹ and μmol h⁻¹, in order to enable readers to compare our HER value with other studies more easily. With regards to the reviewer's comment about the mismatched values, we appreciate their attention to detail and apologize for the mistake in the abstract where the HER value was erroneously written as (0.279 mol h⁻¹). The correct value is 279 μmol h⁻¹.

Comment 12. Figure 2b) is cut-off just below 900 nm which means that the full absorption onset is not visible.

Response: We appreciate the reviewer for bringing to our attention the issue of the unclear absorption onset of the materials in Figure 2b. We concur with the reviewer's observation and have therefore made adjustments to the figure to enhance its clarity. The modified figure has been incorporated into the revised manuscript.

Figure 2: (a) UV–Vis absorption spectra of the all-polymer dots in water solutions.

Comment 13. One significant issue that I have with this work is that figure S26 is shown to vaguely indicate that the reaction is metal free. Palladium will be present in the materials (and should have been quantified) which will act as the proton reduction site. Without significant additional (experimental) evidence of intermediates I cannot see this adding much to the manuscript.

Response: We would like to express our gratitude to the reviewer for their comment. We concur with the reviewer's assessment that we were unable to identify the active site on the polymer photocatalyst in the presence of Pd residue, which serves as the proton reduction site. As per the reviewer's suggestion, we have quantified the Pd residue through Inductively Coupled Plasma Optical Emission spectroscopy (ICP-OES), and the results are presented in **Table S3** of the supplementary information. The ICP-OES results show that the Pd content (wt%) ranges from 0.204 to 0.076, which is sufficient to play an effective role for H₂ evolution^{1, 2}. Therefore, we agree with the reviewer that Figure S26 is only applicable to metal-free photocatalysts, and we will exclude it from the supplementary information.

1. Sachs M, *et al.* Tracking Charge Transfer to Residual Metal Clusters in Conjugated Polymers for Photocatalytic Hydrogen Evolution. *J. Am. Chem. Soc.* **142**, 14574-14587 (2020).
2. Kosco J, *et al.* The Effect of Residual Palladium Catalyst Contamination on the Photocatalytic Hydrogen Evolution Activity of Conjugated Polymers. *Adv. Energy Mater.* **8**, 1802181 (2018).

Table S3: The residual Pd contents were determined by ICP-MS

Polymer	Pd content (wt%)
PITIC-Ph	0.102
PITIC-Th	0.204
PITIC-ThF	0.076
PBTIC-Ph	0.156

PBTIC-Th	0.089
PBTIC-ThF	0.092

Comment 14. How much THF remains in the dispersion used for the photocatalytic experiments? Does this add to the activity?

Response: We would like to extend our gratitude to the reviewer for their perceptive inquiry. With regards to our Pdot preparation method, we are confident that no traces of THF remain in the PDOT solution. To conduct the H₂ evolution test, we prepared individual 10 mL samples for each measurement, with each sample being prepared separately from the others. The concentration of polymer stock used for Pdot solution preparation was 1mg/1mL THF, which means that only a small amount of THF was used for each sample. However, to ensure that any remaining traces of THF were eliminated, we subjected the Pdot solution to a 90-minutes heating process at a temperature of 150 °C on a hotplate. This process resulted in an internal temperature of approximately 100 °C, which was sufficient to remove any residual THF. Our experimental results demonstrate that this rigorous heating process has successfully produced a Pdot solution that is entirely free of THF residue.

Comment 15. The models used for the HOMO/LUMO predictions are potentially too small. The authors should show at least for one of the materials that no changes occur upon adding additional monomer units.

Response: We would like to express our appreciation to the reviewer for their insightful comment regarding the models employed for HOMO/LUMO predictions in our manuscript. To address this point, we conducted additional DFT calculations to investigate the effect of incorporating monomer units to the PITIC-ThF, PITIC-Th, and PITIC-Ph structures, as illustrated in **Figure (S32, S33, and S34)**. Then we add the following sentence for the revised manuscript page 15.

“For the PITIC-ThF polymer, we observed some change occur upon adding additional monomer units (**Figure S34**). Where the HOMOs are not entirely localized on one monomer but instead partially extend to the second monomer. Similarly, the LUMOs also partially extend to the second monomer. This indicates that there is some degree of delocalization through the conjugated chain, though not to the same extent as observed in the PITIC-Th and PITIC-Ph polymers. The underlying reason for this behavior can be attributed to the favorable small dihedral angle of the ThF linker, which facilitates delocalization between the repeated acceptors and along the conjugated chain.”

Figure S32: Electron orbital distributions of the molecular orbitals of the PITIC-Ph polymer consist of **Two** monomer moieties.

Figure S33: Electron orbital distributions of the molecular orbitals of the PITIC-Th polymer consist of **Two** monomer moieties.

Figure S34: Electron orbital distributions of the molecular orbitals of the PITIC-ThF polymer consist of **Two** monomer moieties.

Our results demonstrate that the HOMOs of the PITIC-Th and PITIC-Ph polymers exhibit some degree of delocalization over the conjugated systems of one monomer (Figure S32, and S33). Conversely, the LUMOs are mainly localized over the IC moiety for the second monomer. Which compatible with the HOMO and LUMO calculated using one monomer. For the PITIC-ThF polymer, we observed some change occur upon adding additional monomer units. Where the HOMOs are not entirely localized on one monomer but instead partially extend to the second monomer. Similarly, the LUMOs also partially extend to the second monomer. This indicates that there is some degree of delocalization through the conjugated chain, though not to the same extent as observed in the PITIC-Th and PITIC-Ph polymers. The underlying reason for this behavior can be attributed to the favorable small dihedral angle of the

ThF linker, which facilitates delocalization between the repeated acceptors and along the conjugated chain.

Comment 16. How are the PL spectra in figure S27 comparable given that nanoparticles are present that scatter?

Response: We extend our sincere appreciation to the reviewer for their valuable input. To evaluate the impact of nanoparticle scattering on the photoluminescence (PL) of each sample, we employed the dynamic light scattering (DLS) technique to measure the particle size distribution for all polymer nanoparticle samples. As shown in **Figure S43**, the particle size distribution of all samples was found to be within a similar range of 5-50 nm. This indicates that the difference in PL intensity observed among different polymer samples (as presented in **Figure S36**) is not likely to be caused by differences in particle size. Instead, it can be attributed to charge separation, which results from the variation in the linker structure within the polymer compositions.

The difference in dihedral angles between the acceptor and linker groups in each polymer gives rise to charge separation. These linker groups affect the charge delocalization between the repeated acceptors (IC moiety) along the polymer chains and can modify the charge recombination processes within the polymer structures. As a result, although the particle size distribution may be similar, each polymer may display distinctive PL characteristics. Consequently, our findings indicate that the linker structure plays a crucial role in determining the PL properties of the polymer nanoparticles.

Figure S42: Particle size distributions measured by dynamic light scattering (DLS) of all presented polymers nanoparticles in water.

As shown in **Figure S43**, the particle size distribution of all samples was found to be within a similar range of 5-50 nm. This indicates that the difference in PL intensity observed among different polymer samples (as presented in **Figure S36**) is not likely to be caused by differences in particle size. Instead, it can be attributed to charge separation, which results from the variation in the linker structure within the polymer compositions.

Figure S36: Steady-state photoluminescence (PL) spectra of all presented polymers nanoparticles in water.

Comment 17. Details relating to the electrochemical impedance spectroscopy measurements are missing (how were these experiments set up? What circuit configuration was used?). Where the nanomaterials studied? If films were used how do the authors ensure that they are representative of the nanomaterials? How was the contact between the material and electrode characterized given that this is an interface which does not exist in photocatalysis? The discussion is also not very detailed and should provide values for resistance, capacitance, Warburg impedance, and the charge carrier transfer resistance rather than making handwavy statements.

Response: We thank the reviewer for his careful reading and valuable comments. We would answer this question point by point as the following.

- how were these experiments set up?

All the details relating to the electrochemical impedance spectroscopy measurements were mentioned in our revised manuscript as follow:

“The Zahner Zennium E workstation, which featured a three-electrode cell consisting of a Pt wire counter electrode, an Ag/AgCl reference electrode (3M NaCl), and a fluorine-doped tin oxide (FTO) glass working electrode, was utilized to perform Electrochemical Impedance Spectroscopy (EIS) measurements. An active area of 1 cm² FTO glass was drop-casted with 0.5 mL of a Pdot solution (1 mg/mL) at a temperature of 70-80°C. EIS spectra were collected at a voltage of 1.5 V_{Ag/AgCl} in 0.5 M Na₂SO₄ electrolyte solution with an amplitude of 20 mV and a frequency range of 1 Hz to 100 kHz, while the sample was illuminated with a white light-emitting diode (LED) PAR38 lamp (20 W, 6500 K, Zenaro Lighting; $\lambda > 420$ nm).”

- What circuit configuration was used?

An equivalent circuit model was used to fit the Nyquist plot, composed of solution resistance (R_s), double layer capacitor (C_{dl}), and the charge transfer resistance (R_{ct}), as shown in the following schematic draw.

- Where the nanomaterials studied? If films were used how do the authors ensure that they are representative of the nanomaterials?

As previously described, we utilized a material film for the purpose of measuring EIS, which was produced by drop casting polymer nanoparticles onto the FTO electrode. However, a lingering uncertainty remained as to whether the nanoparticle structure of the polymers had uniformly deposited or aggregated after drop casting onto the FTO substrate. This uncertainty stems from the fact that there are multiple factors that can influence the uniformity of the polymer nanoparticle film, including the concentration of the polymer nanoparticles, heating rate during drop casting, and other factors. To clarify the uniformity of the polymer nanoparticle film, further investigation is warranted. For instance, the concentration of the polymer nanoparticles can impact the resulting film's uniformity as reported by Hyunjin Jo et al (Macromol. Mater. Eng. 2016, 301, 530–534). Increasing the concentration may lead to aggregation or uneven deposition, while decreasing the concentration may lead to insufficient coverage of the FTO electrode. So, it is expected to be a wide and interesting point in the future for full studying.

- How was the contact between the material and electrode characterized given that this is an interface which does not exist in photocatalysis?

Our primary objective for conducting Electrochemical Impedance Spectroscopy (EIS) measurements is to investigate some of the intrinsic properties of the polymer, rather than those pertaining to the photocatalysis system. EIS is a powerful analytical technique that provides valuable insights into the charge transfer processes occurring at the polymer-liquid interface.

Through EIS, we are able to determine the charge transfer resistance value for each photocatalyst, which provides an indication of the possibility of transfer charges between the photocatalyst and liquid interface. For our case, our results indicate that the charge transfer is significantly affected by the use or absence of surfactant. Surfactants, due to their non-conductive properties, may impede the charge transfer between the semiconducting polymer (such as PITIC-ThF) and liquid electrolyte. This observation suggests that a similar effect may occur in the photocatalysis system.

In the photocatalysis system, a non-conductive surfactant that covers the polymer photocatalyst from the outside may hinder or reduce the charge transfer between the photocatalyst and reactants in the solution, such as water or platinum cocatalyst. This is because the surfactant acts as a barrier, preventing the reactants from reaching the photocatalyst surface and inhibiting the charge transfer process. As a result, the photocatalytic activity may be significantly reduced.

The literature also provides several examples where EIS measurements have been used to explain the properties of photocatalysts that affect their behavior in the photocatalysis system.^{3,4}

3. Jian J, *et al.* Embedding laser generated nanocrystals in BiVO(4) photoanode for efficient photoelectrochemical water splitting. *Nat. Commun.* **10**, 2609 (2019).
4. Verma P, *et al.* Charge-transfer regulated visible light driven photocatalytic H(2) production and CO(2) reduction in tetrathiafulvalene based coordination polymer gel. *Nat. Commu.n* **12**, 7313 (2021).

- The discussion is also not very detailed and should provide values for resistance, capacitance, Warburg impedance, and the charge carrier transfer resistance rather than making handwavy statements.

The results of the electrochemical impedance spectroscopy measurements are discussed in more detail in the results and discussion section and fitted parameter values of equivalent circuit based on impedance spectra of all the prepared catalysts were also provided. We've mentioned that in our revised manuscript and supplementary information as follow:

“According to the results obtained from the electrochemical impedance spectroscopy (EIS) technique (**Figure 7b**), the presence of surfactants is found to increase the charge transfer resistance. The EIS Nyquist plot clearly shows that the semicircle diameter for PS-PEG-COOH/PITIC-ThF Pdots is larger than that of PITIC-ThF Pdots, indicating sluggish charge transfer kinetics. Furthermore, the fitting of the Nyquist plot by means of an equivalent circuit model reveals that the PITIC-ThF Pdots exhibit a much lower charge transfer resistance (R_{ct}) of 48.06 Ω compared to PS-PEG-COOH/PITIC-ThF Pdots ($R_{ct} = 73.74 \Omega$) (see **table S6**). This implies that the absence of surfactants facilitates electron transfer between the polymer electrode and electrolyte solution, while the presence of surfactants increases the resistance and impairs electron transfer. The most plausible explanation for this phenomenon is the non-conductive nature of the surfactant that acts as a barrier when it covers the Pdot particles, thus hindering electron transfer from the semiconducting polymer (PITIC-ThF) to the electrolyte solution.⁵ This behavior is also anticipated to occur during the photocatalysis process, where covering PITIC-ThF with non-conductive surfactants can impede charge transfer from the photocatalyst to the reactants or Pt cocatalyst in the solution.

5. Elsayed MH, *et al.* Hydrophobic and Hydrophilic Conjugated Polymer Dots as Binary Photocatalysts for Enhanced Visible-Light-Driven Hydrogen Evolution through Forster Resonance Energy Transfer. *ACS Appl. Mater. Interfaces* **13**, 56554-56565 (2021).

Figure 7. (a) Effect of PS-PEGCOOH surfactant on the photocatalytic hydrogen production activity of the PITIC-ThF Pdots. (b) Electrochemical impedance spectroscopy (EIS) of the PITIC-ThF Pdots (with and without a surfactant). (d) Schematic diagram of the PITIC-ThF Pdots (with and without a surfactant) presenting the effect of the surfactant on the charge transfer between the Pdots and Pt cocatalyst.

Figure S27: (a) and (b) Photoluminescence emission spectra of PITIC-X and PB TIC-X based Pdots,

respectively. (c) and (d) The electrochemical impedance spectroscopy (EIS) of PITIC-X and PBTIC-X based Pdots, respectively. (e) Photocurrent response of presented polymers. (f) Time-resolved decay traces of pristine PITIC-ThF and PBTIC-ThF as solutions in water (excitation wavelength, 600 nm; emission wavelength, 750 and 800 nm, respectively).

Figure S41: (a) Effect of Triton surfactant on the photocatalytic hydrogen production activity of the PITIC-ThF Pdots. (b) Electrochemical impedance spectroscopy (EIS) of the PITIC-ThF Pdots (with and without Triton surfactant).

Table S6: The equivalent circuit fitted results of EIS data in Fig. 6c.

Electrodes	R _s (Ω)	R _{ct} (Ω)	C _{dl} (F)
PITIC-ThF Pdot (without surfactant)	18.18	48.06	9.862 x 10 ⁻⁶
PITIC-ThF PS-PEG-COOH Pdot	17.2	73.74	9.819 x 10 ⁻⁶

Table S7: The equivalent circuit fitted results of EIS data in Fig. S--.

Electrodes	R _s (Ω)	R _{ct} (Ω)	C _{dl} (F)
PITIC-ThF Pdot (without surfactant)	18.18	48.06	9.862 x 10 ⁻⁶
PITIC-ThF Triton Pdot	17.68	78.28	9.244 x 10 ⁻⁶

Table S8: The equivalent circuit fitted results of EIS data in Fig. S27c, and S27d.

Electrodes	R _s (Ω)	R _{ct} (Ω)	C _{dl} (F)
PBTIC-Ph Pdot	16.33	71.41	11.56 x 10 ⁻⁶
PBTIC-Th Pdot	18.27	52.33	9.414 x 10 ⁻⁶
PBTIC-ThF Pdot	14.91	42.38	9.54 x 10 ⁻⁶
PITIC-Ph Pdot	17.02	80.99	9.833 x 10 ⁻⁶
PITIC-Th Pdot	16.4	62.14	7.4 x 10 ⁻⁶
PITIC-ThF Pdot	18.18	48.06	9.862 x 10 ⁻⁶

Comment 18. The ESI is lacking many details that would allow other researcher to reproduce the results.

Response: We extend our appreciation to the reviewer for their kind feedback. We have conducted a thorough examination of the electrochemical impedance spectroscopy measurements and have comprehensively documented all relevant information in the Experimental section. Additionally, we have elaborated on the EIS results in greater detail in the Results and Discussion section, as elucidated in our response to the previous reviewer's comment (Comment 17).

- a) The synthesis section only provides proton NMR. For any synthetic work this is not acceptable and as a bare minimum CHN analysis and mass spec data has to be provided to show that the intermediates and monomers are pure and of the suggested composition.

Response: We express our gratitude to the reviewer for providing valuable feedback. In response, we have conducted CHN analysis and mass spectra analysis on the monomers (IC-Br, BTIC-Br, and ITIC-Br) and incorporated the data in the experimental section of the monomer synthesis. Our findings reveal that the HR-FD-MS and CHN analysis values of IC-Br, BTIC-Br, and ITIC-Br monomers were in concordance with their theoretical molecular weight, indicating a high degree of purity of these compounds. The data has been included in the supplementary information and inserted into the experimental section of the monomer synthesis as following.

“Synthesis of compound 5 (IC-Br) (2-(5(6)-bromo-3-oxo-2,3-dihydro-1H-inden-1-ylidene) malononitrile). Compound 3 (2.33 g, 10.35 mmol) and malononitrile (1.37 g, 20.7 mmol) were mixed in 40 mL ethanol in a 250 mL single neck round bottom flask and stirred for 30 min at room temperature. Then, anhydrous sodium acetate (1.28 g, 15.52 mmol) was added to the reaction, and the mixture was stirred at room temperature for 2 h. After the reaction 40 mL water was added, and the mixture was stirred at room temperature for half an hour. Then, concentrated HCl was dropped into the mixture to acidify the mixture with pH = 2. The precipitate was filtered and washed with water many times. The crude product was further purified by flash column chromatography to afford title compound 3 as yellow solid (0.38 g, yield: 29.4%). ¹H NMR (500 MHz, CDCl₃) δ 8.75 (s, 1H), 8.49 (d, 1H), 8.1 (s, 1H), 7.97 (m, 1H), 7.82 (d, 1H), 3.72 (d, 2H). HR-FD-MS: m/z: 273.9590 for C₁₂H₅BrN₂O. Elemental analysis found C 52.31%, N 10.06 %, H 2.29 %, O 6.06 %.

Synthesis of compound 7 (Br-ITIC-Br). Compound 5 (558 mg, 2.04 mmol) and Compound 6 (400 mg, 0.38 mmol) were added to a 250 ml two-necked round bottom flask, after being rinsed with a mild stream of nitrogen for ten minutes, then anhydrous chloroform (100 mL) was added to the flask. Finally, pyridine (2 mL) was added to the reaction, the mixture turned green gradually. Then, the reaction was placed in an oil bath at 65 °C stirred and refluxed for 12 hours. After the reaction was completed, it was cooled to room temperature. The crude product was poured into methanol (400 mL), then the precipitate was filtered and purified by flash column (DCM/Hexane (1:1)). ¹H NMR (500 MHz, CDCl₃) δ 8.85 (d, 1H), 8.79 (dd, 0.5H), 8.52 (dd, 0.5H), 8.21 (d, 1H), 7.99 (dd, 0.5H), 7.85-7.82 (m, 1H), 7.74 (dd, 0.5H), 7.62 (m, 1H), 7.19-7.10 (dd, 8H), 2.56-2.53 (m, 4H), 1.60-1.52 (m, 4H), 1.32 (dq, 12H), 0.84 (t, 6H). HR-FD-MS: m/z: 1582.3536 for C₉₄H₈₀Br₂N₄O₂S₄. Elemental analysis found C 71.55 %, N 3.55 %, H 5.19 %, O 2.24 %, S 6.86%.

Synthesis of compound 10 (Br-BTIC-Br). Compound 5 (558 mg, 2.04 mmol) and Compound 9 (400 mg, 0.38 mmol) were added to a 250 ml two-necked round bottom flask, after being rinsed with a mild stream of nitrogen for ten minutes, then anhydrous chloroform (100 mL) was added to the flask. Finally,

pyridine (2 mL) was added to the reaction, the mixture turned green gradually. Then, the reaction was placed in an oil bath at 65 °C stirred and refluxed for 12 hours. After the reaction was completed, it was cooled to room temperature. The crude product was poured into methanol (400 mL), then the precipitate was filtered and purified by flash column (DCM/Hexane (1:1)). ¹H NMR (500 MHz, CDCl₃) δ 9.14 (s, 1H), 8.68 (d, J = 5.7 Hz, 0.5H), 7.96 (s, 0.5H), 7.83 – 7.65 (m, 2H), 4.90 – 4.67 (m, 4H), 3.22 (t, J = 7.6 Hz, 4H), 2.10 – 2.00 (m, 2H), 1.90-1.80 (m, 2H), 1.41 – 0.92 (m, 52H), 0.81-0.60 (m, 20H), 0.68 (m, 6H). HR-FD-MS: m/z: 1534.4006 for C₈₂H₈Br₂N₈O₂S₅. Elemental analysis found C 64.67 %, N 6.71 %, H 5.95 %, O 3.22 %, S 9.55%.

Figure S11: High resolution mass spectra of IC-Br monomer. HR-FD-MS: m/z: 273.9590. The chemical formula and theoretical molecular weight of IC-Br are C₁₂H₅BrN₂O and 273.09, respectively.

Figure S12: High resolution mass spectra of Br-ITIC-Br monomer. HR-FD-MS: m/z: 1582.3536. The chemical formula and theoretical molecular weight of Br-ITIC-Br are C₉₄H₈₀Br₂N₄O₂S₄ and 1585.75, respectively.

Figure S13: High resolution mass spectra of Br-BTIC-Br monomer. HR-FD-MS: m/z : 1534.4006. The chemical formula and theoretical molecular weight of Br-BTIC-Br are $C_{82}H_{88}Br_2N_8O_2S_5$ and 1537.77, respectively.

- b) For an ESI the use of abbreviations is excessive and full names should be used in the procedures to make it clearer to the reader what was used without constantly cross-referencing.

Response: Thanks for the editor notice. I agree with the editor's comment that when creating an EIS, it is better to use full names instead of excessive abbreviations in order to ensure clarity and avoid confusion for the reader. So, we use the full name of EIS (Electrochemical Impedance Spectroscopy) in the description of EIS experiments and discussion as the following and modified in the manuscript.

“The Zahner Zennium E workstation, which featured a three-electrode cell consisting of a Pt wire counter electrode, an Ag/AgCl reference electrode (3M NaCl), and a fluorine-doped tin oxide (FTO) glass working electrode, was utilized to perform Electrochemical Impedance Spectroscopy (EIS) measurements. An active area of 1 cm^2 FTO glass was drop-casted with 0.5 mL of a Pdot solution (1 mg/mL) at a temperature of $70\text{-}80^\circ\text{C}$. EIS spectra were collected at a voltage of $1.5\text{ V}_{\text{Ag/AgCl}}$ in $0.5\text{ M Na}_2\text{SO}_4$ electrolyte solution with an amplitude of 20 mV and a frequency range of 1 Hz to 100 kHz, while the sample was illuminated with a white light-emitting diode (LED) PAR38 lamp (20 W, 6500 K, Zenaro Lighting; $\lambda > 420\text{ nm}$).”

- c) The synthetic procedures for the polymerisations also lack detail. What amounts and stoichiometry was used? For example, it states Monomer ‘Br-ITIC-Br or Br-BTIC-Br (152 mg 0.1 mmol)’ which indicates only the amounts and stoichiometry of the second monomer.

Response: Thanks for the precious notice of the reviewer. We modified the description of synthetic procedures of the polymerisations with more details to be the following:

“**Synthesis of PITIC-Ph and PBTIC-Ph polymers:** PITIC-Ph and PBTIC-Ph were prepared by Suzuki–Miyaura coupling polymerization. Monomer Br-ITIC-Br (158.5 mg 0.1 mmol) or Br-BTIC-Br (153.2 mg 0.1 mmol), respectively, monomer B-Ph-B (33 mg, 0.1 mmol), Na_2CO_3 (79 mg, 0.75 mmol),

tetra-n-butylammonium bromide (1.6 mg, 0.005 mmol), and Pd(PPh₃)₄ (9.0 mg, 0.008 mmol), toluene (10 mL), and water (2.5 mL) were injected into a sealed tube (Recently, We have opted for the refluxing technique in a round bottom flask (more safety) over using a sealed tube which give the same results). The mixture was degassed by bubbling with N₂ for 30 min and then heated at 120 °C for 72 h. After cooling to room temperature, bromobenzene was added and then the sealed tube was heated at 120 °C for 6 h, followed by the addition of phenylboronic acid and heating at 120 °C for another 6 h. The mixture was cooled to room temperature and poured into MeOH. The precipitate was collected through membrane filtration. Purification of the polymer was performed through Soxhlet extraction with MeOH and hexane. Finally, the polymer was dissolved in hot CHCl₃, concentrated, and then precipitated in MeOH. The polymer was collected and dried under vacuum. The PITIC-Ph and PBTIC-Ph were isolated as powders in 63% and 57% yield, respectively.

Synthesis of PITIC-Th and PBTIC-Th polymers: PITIC-Th and PBTIC-Th were prepared by Stille coupling polymerization. Monomer Br-ITIC-Br (158.5 mg 0.1 mmol) or Br-BTIC-Br (153.2 mg 0.1 mmol), respectively, monomer Sn-Th-Sn (41 mg, 0.1 mmol), Pd(PPh₃)₄ (9.0 mg, 0.008 mmol), and anhydrous toluene (10 mL) were added to a sealed tube (Recently, We have opted for the refluxing technique in a round bottom flask (more safety) over using a sealed tube which give the same results). The mixture was degassed by bubbling with N₂ for 30 min and then stirred at 100 °C for 24 h. After cooling to room temperature, the mixture was poured into MeOH. The precipitate was collected through membrane filtration. Purification of the polymer was performed through Soxhlet extraction with MeOH and hexane. Finally, the polymer was dissolved in hot CHCl₃, concentrated, and then precipitated in MeOH. The polymer was collected and dried under vacuum. The PITIC-Th and PBTIC-Th were isolated as powders in 51% and 54% yield, respectively.

Synthesis of PITIC-ThF and PBTIC-ThF polymer: PITIC-Th and PBTIC-ThF were prepared by the same synthetic method of PITIC-Th mentioned above. Monomer Br-ITIC-Br (158.5 mg 0.1 mmol) or Br-BTIC-Br (153.2 mg 0.1 mmol), respectively, monomer Sn-ThF-Sn (44.6 mg, 0.1 mmol), Pd(PPh₃)₄ (9.0 mg, 0.008 mmol), and anhydrous toluene (10 mL) were added to a sealed tube (Recently, We have opted for the refluxing technique in a round bottom flask (more safety) over using a sealed tube which give the same results). Then follow the same procedure above. The PITIC-ThF and PBTIC-ThF were isolated as powders in 52% and 55% yield, respectively.

d) The procedure states that a ‘sealed tube’ was used. Is this the case? Given the temperature above the boiling point of both water and toluene this should actually be noted to be of risk to others in the text. I have seen many student producing bombs which caused terrible accidents, some with injuries that could have been prevented.

Response: Thanks for reviewer for bringing the reviewer's notification to our attention. We completely agree with their concerns regarding the dangers of conducting polymerization experiments in a sealed tube. Our lab recently has switched to using the refluxing technique in a round bottom flask for all of our polymerization experiments and has completely stopped using sealed tubes.

In response to the reviewer's suggestion, we have updated our polymerization procedure to reflect this change and have included a note for readers to be aware of the potential dangers of using sealed tubes. Please refer to the polymerization method provided in the answer to comment 18 (c).

d) For the polymerisations no yields are given.

Response: We express our appreciation to the reviewer for their significant input. The polymerization yields were determined and included with the methods of polymerization as described below.

“PITIC-Ph and PBTIC-Ph were obtained as powders with a yield of 63% and 57%, respectively.”

“PITIC-Th and PBTIC-Th were obtained as powders with a yield of 51% and 54%, respectively.”

“PITIC-ThF and PBTIC-ThF were obtained as powders with a yield of 52% and 51%, respectively.”

Comment 19. Why do the materials do not fully decompose at temperatures exceeding 500 deg C?

Response: We extend our gratitude to the reviewer for their valuable feedback. It has come to our attention that incomplete decomposition of the materials at temperatures exceeding 500 degrees Celsius may be due to the fact that our TGA measurements were performed under N₂ inert conditions and in the absence of O₂. The presence of inert N₂ atmosphere can lead to the polymer converting into a thermodynamically stable char structure, which exhibits a considerably slower degradation rate at high temperatures, as shown in Figure R1 and R2. This finding sheds light on the importance of carefully considering the choice of atmosphere during TGA measurements, as it can significantly impact the decomposition behavior of the materials under study.^{6,7}

6. Iacono ST, Ewald D, Sankhe A, Rettenbacher A, Smith DW. Sulfonated Fluorovinylene Aromatic Ether Polymers for Proton Exchange Membranes. *High Performance Polymers* **19**, 581-591 (2008).

7. Ramgobin A, Fontaine G, Bourbigot S. A Case Study of Polyether Ether Ketone (I): Investigating the Thermal and Fire Behavior of a High-Performance Material. *Polymers (Basel)* **12**, 1789 (2020).

Figure R1. TGA and DTG plots of polyether ether ketone (PEEK) when heated at 10 °C/min under pyrolytic conditions (black, square), 2% oxygen (red, circle), 4% oxygen (blue, upwards triangle), 8% oxygen (magenta, downwards triangle), 12% oxygen (green, diamond) and in synthetic air (brown, left-

pointing triangle). Reprinted from ref (7)⁷.

Figure R2. TGA trace of polymer 7 in nitrogen and air. Reprinted from ref (6)⁶.

Reviewer #2 (Remarks to the Author):

This work reports polymers with absorption in NIR region for photocatalytic hydrogen production and aims to study the charge separation and recombination processes. The synthesized polymers are interesting and the photocatalytic performance is good. However, there are much data which can not well support the discussion and conclusion.

1. Impedance results are used to explain that PS-PEG-COOH/PITIC-ThF Pdots has large resistance than PITIC-ThF Pdot because the surfactant hinders the interaction between polymers. This is not correct. The impedance mainly measured the resistance from interface charge transfer. The Pdots with surfactant of course have long hydrophilic arms which will weaken the interaction with the electrode. But it does not mean the inner interaction between polymers is weak.

Response: The commentary provided by the reviewer is acknowledged with gratitude. Our concurrence with the reviewer's assertion that the electrochemical impedance spectroscopy (EIS) principally measures the charge transfer resistance at the polymer electrode and electrolyte solution interface is affirmed. To elucidate the charge transfer resistance, we employed an equivalent circuit model to fit the EIS data, thereby determining the solution resistance (R_s), double layer capacitor (C_{dl}), and charge transfer resistance (R_{ct}). As a result, our manuscript has been revised to incorporate these findings into our discussion. Also we add a table contain all EIS fitting constants with modifying figure 6c and S29 as the following.

“According to the results obtained from the electrochemical impedance spectroscopy (EIS) technique (Figure 7b), the presence of surfactants is found to increase the charge transfer resistance. The EIS Nyquist plot clearly shows that the semicircle diameter for PS-PEG-COOH/PITIC-ThF Pdots is larger than that of PITIC-ThF Pdots, indicating sluggish charge transfer kinetics. Furthermore, the fitting of the Nyquist plot by means of an equivalent circuit model reveals that the PITIC-ThF Pdots exhibit a much lower charge transfer resistance (R_{ct}) of 48.06Ω compared to PS-PEG-COOH/PITIC-ThF Pdots ($R_{ct} = 73.74 \Omega$) (see table S6). This implies that the absence of surfactants facilitates electron

transfer between the polymer electrode and electrolyte solution, while the presence of surfactants increases the resistance and impairs electron transfer. The most plausible explanation for this phenomenon is the non-conductive nature of the surfactant that acts as a barrier when it covers the Pdot particles, thus hindering electron transfer from the semiconducting polymer (PITIC-ThF) to the electrolyte solution.⁵ This behavior is also anticipated to occur during the photocatalysis process, where covering PITIC-ThF with non-conductive surfactants can impede charge transfer from the photocatalyst to the reactants or Pt cocatalyst in the solution.”

5. Elsayed MH, *et al.* Hydrophobic and Hydrophilic Conjugated Polymer Dots as Binary Photocatalysts for Enhanced Visible-Light-Driven Hydrogen Evolution through Forster Resonance Energy Transfer. *ACS Appl. Mater. Interfaces* **13**, 56554-56565 (2021).

Figure 7. (a) Effect of PS-PEGCOOH surfactant on the photocatalytic hydrogen production activity of the PITIC-ThF Pdots. (b) Electrochemical impedance spectroscopy (EIS) of the PITIC-ThF Pdots (with and without a surfactant). (c) Schematic diagram of the PITIC-ThF Pdots (with and without a surfactant) presenting the effect of the surfactant on the charge transfer between the Pdots and Pt cocatalyst.

Figure S42: Figure S29. (a) Effect of Triton surfactant on the photocatalytic hydrogen production activity of the PITIC-ThF Pdots. (b) Electrochemical impedance spectroscopy (EIS) of the PITIC-ThF Pdots (with and without Triton surfactant).

Table S6: The equivalent circuit fitted results of EIS data in Fig. 7c.

Electrodes	R_s (Ω)	R_{ct} (Ω)	C_{dl} (F)
PITIC-ThF Pdote (without surfactant)	18.18	48.06	9.862×10^{-6}
PITIC-ThF PS-PEG-COOH Pdote	17.2	73.74	9.819×10^{-6}

Table S7: The equivalent circuit fitted results of EIS data in Fig. S42b.

Electrodes	R_s (Ω)	R_{ct} (Ω)	C_{dl} (F)
PITIC-ThF Pdote (without surfactant)	18.18	48.06	9.862×10^{-6}
PITIC-ThF Triton Pdote	17.68	78.28	9.244×10^{-6}

2. Figure 3 g and h provide misleading information, one cannot simply compare EQE and hydrogen generation rate per gram catalyst. Unless all systems used same amount catalyst in a certain volume.

Response: Thanks for the valuable feedback of the reviewer on the direct comparison of AQYs/EQEs. We acknowledge that experimental factors can influence the measurement of HER, making it challenging to directly compare results across different studies. To provide transparency and clarity, we collected and organized all the conditions that revealed in other articles. Moreover, as recommended, we summarized the AQYs/EQEs of different material categories to offer a more comprehensive assessment of the new materials' performance. Table S4 presents a comparison of various polymer photocatalysts, including their respective photocatalytic reaction conditions. In addition, our photocatalyst is compared to other organic and inorganic photocatalysts in terms of their photocatalytic activity when exposed to both visible and near-infrared (NIR) light (Table S5).

Table S4. Comparative studies of our developed polymer photocatalyst versus other polymer photocatalysts in terms of HER, AQY% with the reaction conditions and light source.

Polymer ^a	Conditions	Light Source	HER (mmol h ⁻¹ g ⁻¹)	AQY % at wavelengths (nm)	References
PFBT-Pdots	0.075 mg in 3 mL of 0.2M ascorbic acid.	A LED lamp ($\lambda > 420$ nm)	8.3	0.5 at 420 nm	Angew. Chem. Int. Ed. 2016, 55, 12306–12310
PFODTBT Pdot	0.075 mg in 3 mL of 0.2M ascorbic acid.	A LED lamp ($\lambda > 420$ nm)	50.0	0.9, 0.3, 0.6, and 0.3 at 420, 500, 550, and 600 nm	Energy Environ. Sci., 2017,10, 1372-1376
PFTFQ-PtPy15	1 mg in 10 mL Water/ 20 vol% TEA	A LED lamp ($\lambda > 420$ nm)	12.7	0.4 at 500 nm	ACS Catal. 2018, 8, 7766–7772
F8T2 Pdots/g-C3N4	20 mg in (90 mL Water + 10 mL TEOA)	300 W Xenon lamp ($\lambda > 400$ nm)	0.93	5.7, 2.8, and 0.8 at 420, 500, and 550 nm	J. Mater. Chem. A, 2019, 7, 303-311
HE-CP10-Dots	20 mg + 55 mL water + ascorbic acid (1.76 g)	300 W Xenon lamp ($\lambda > 420$ nm)	0.84	0.9 at 500 nm	Macromolecules 2019, 52, 11, 4376–4384
F8DTBT Pdots/CNN S	20 mg in (90 mL Water + 10 mL TEOA)	300 W Xenon lamp ($\lambda > 400$ nm)	0.181	3.4, 0.4, 0.2, and 0.5 at 420, 500, 550, and 600 nm	Appl. Catal. B: Environ. 2019, 259, 118067
PFN-Br	2.5 mg in (5 mL TEOA + 45 mL water)	300-W Xe lamp ($\lambda > 300$ nm)	0.68	0.12, 0.40, 0.44 and 0.19 at 550, 600, 650 and 700 nm	Nano Energy, 2019, 60, 775–783
PBDTBT-7EO (3.0 wt% Pt)	2.5 mg in 50 mL of AA solution (0.2 M)	300-W Xe lamp ($\lambda > 300$ nm)	15.9	0.13, 0.14, 0.25, and 0.30 at 420, 500, 550, and 600 nm	iScience, 2019, 13, 33–42.
PFTBTA-PtPy	1 mg in 10 mL Water/ 20 vol% TEA	A LED lamp ($\lambda > 420$ nm)	7.34	0.5 at 420 nm	Appl. Catal. B: Environ. 2020, 268, 118436
PNBTBr Pdots/CNN S	20 mg in (90 mL Water + 10 mL TEOA)	300 W Xe lamp ($\lambda > 400$ nm)	1.2	7.71, 2.5, 2.0 at 420, 500, and 550 nm	Appl. Catal. B: Environ. 2020, 270, 118852
PTB7-Th/EH-IDTBR	2 mg in 20 mL 0.2M AA solution	300-W Xe lamp	28.13	2.0, 2.3, 4.3, 5.6, and 6.2 at 420, 500, 620, 660,	Nat. Mater. 2020, 19, 559–565

Pdot				and 700 nm	
PyDTDO-3 (w/o Pt)	1.0 M AA solution / 10 vol% DMF	10 mg in (90 mL of 1M AA + 10 mL DMF)	16.32	3.70, 3.68, 3.93, and 2.30 at 420, 500, 550, and 600 nm	Chem. Sci., 2021,12, 1796-1802.
PS-PEG5-FNP	0.2M AA solution	5 mg in (25 mL ascorbic acid (0.2 M))	37.2	2.5 at 420 nm	Angew. Chem. Int. Ed. 2021, 60, 15590–15597.
PyBS-3	25 mg in (100 mL 0.2M AA solution)	300 W Xenon lamp ($\lambda > 300$ nm)	100.1	29.3 at 420 nm	Adv. Mater. 2021, 33, 2008498.
ZnCoP-F CP	30 mg in (42.5 mL water + 7.5 mL TEOA)	300 W Xe-lamp with a cutoff filter ($\lambda \geq 400$ nm)	2.76	6.92, 5.19, 5.50, 5.78, 3.17, 1.93 at 400, 450, 500, 550, 700, and 760 nm	Adv. Funct. Mater. 2021, 31, 2009819.
D1/D2/ITI C	0.062 mg in (1.5 mL water + 0.5 mL AA 0.8 M)	LED PAR38 lamp ($\lambda > 420$ nm)	60.8	2.2, 4.6, 6.5, 7.1, 6.1, 4.1 at 450, 500, 550, 600, 650, and 700 nm	J. Am. Chem. Soc. 2021, 143, 2875.
gIDTBT:oI DTB R	1 mg in (0.2 M AA (12 mL))	Solar simulator (Asahi Max 303) and an AM1.5g filter	18.5	5.3/1.0/2.9/2.8/0.9% at 400, 440, 620, 660 and 700 nm	Adv. Mater. 2021, 34, 2105007.
PCPDTBS O	2 mg in (1mL NMP + 9 mL ascorbic acid (1M))	300 W Xenon lamp ($\lambda > 350$ nm)	24.6	0.94, 7.77, 8.72, 4.77, 3.74 at 420, 460, 500, 550, and 600 nm	Appl. Catal. B: Environ. 2021, 298, 120577.
PBDTTS-ISO	6 mg in (3mL NMP + 27 mL ascorbic acid (1M))	300 W Xenon lamp ($\lambda > 350$ nm)	97.12	13.5, 16.7, 18.5, and 9.8 at 420, 500, 550, and 600 nm	J. Mater. Chem. A, 2022,10, 6641-6648
PM6:PCBM 2:8	1 mg in (0.2 M AA (12 ml))	Solar simulator (Asahi Max 303) and an AM1.5g filter	73.7	8.7, 8.8, 7.7, 6.6, 2.6 at 400, 470, 560, 620 and 700 nm	Nat. Energy. 2022, 7, 340-351
PBTIC-ThF Pdot	0.1 mg in (10 mL 0.1M ascorbic acid)	A LED lamp ($\lambda > 420$ nm)	269.4	3.9, 3.2, 3.1, 3.9, and 4.7 at 420, 500, 550, 600 and 700 nm	This work
PITIC-ThF Pdot	0.1 mg in (10 mL 0.1M ascorbic acid)	A LED lamp ($\lambda > 420$ nm)	339.7	2.9, 2.7, 2.5, 2.8, and 3.1 at 420, 500, 550, 600 and 700 nm	This work

	acid)				
--	--------------	--	--	--	--

Table S5. Comparative studies of our developed polymer photocatalyst versus other photocatalysts in terms of photocatalytic hydrogen evolution under both visible and NIR light.

Photocatalysts	Conditions	Visible light (>420 nm) (mmol g ⁻¹ h ⁻¹)	NIR light (>780 nm) (μmolg ⁻¹ h ⁻¹)	References
Au/La ₂ Ti ₂ O ₇	1.5 mg in 5 mL (1:4) methanol-H ₂ O solution	0.74	300	Angew. Chem. Int. Ed. 2017, 56, 2064–2068
g-C ₃ N ₄ -Co-K	50 mg in (15 mL TEOA + 85 mL H ₂ O).	0.808	470	J. Colloid Interface Sci. 2020, 561, 719–729
WS ₂ @Cu Hybrids	3 mg + 90 mL H ₂ O + 10 mL Lactic Acid + 1g PEG	64	175 (>750 nm)	Adv. Funct. Mater. 2018, 28, 1804055.
Black Phosphorus / g-C ₃ N ₄	1.5 mg in (1 ml methanol + 4 mL H ₂ O)	0.427	101	J. Am. Chem. Soc. 2017, 139, 13234–13242
Au / La ₂ Ti ₂ O ₇	10 mg in (2 mL methanol + 8 mL H ₂ O)	0.34	180	ACS Catal. 2018, 8, 122–131.
g-C ₃ N ₄ / Chlorin e6	10 mg in (4 mL TEOA + 16 mL H ₂ O)	1.275	312.6	Appl. Catal. B: Environ. 2020, 260, 118137.
CuNi / rGO composite	5 mg in (60 mL H ₂ O + 10 mL lactic acid)	1.787	86	J. Mater. Chem. A, 2017, 5, 22772–22781.
Black Phosphorus / TiO ₂	2 mg in (1 mL methanol + 4 mL H ₂ O)	0.941	200	ACS Catal. 2019, 9, 3618–3626.
H _{0.53} WO ₃ / CdS – Au	30 mg in (10 mL lactic acid + 90 mL H ₂ O)	10	158	J. Mater. Chem. A, 2019, 7, 1076–1082.
C/K-doped RPCN	20 mg in (3 mL TEOA + 27 mL)	1.4	140	Adv. Mater. 2021, 33, 2101455.
PBTIC-ThF Pdot	0.1 mg in (10 mL 0.1M ascorbic acid)	269.4	---	This work
PBTIC-ThF Pdot	5 mg in (10 mL 0.1M ascorbic acid)	35.6	708	This work
PITIC-ThF Pdot	0.1 mg in (10 mL 0.1M ascorbic acid)	339.7	---	This work
PITIC-ThF Pdot	5 mg in (10 mL 0.1M ascorbic acid)	55.8	4045	This work

3. From the title, this work should focus on charge recombination study, however, TAS study is

superficial and does not fully support the conclusion. The conclusion of charge recombination with the order of ThF > Th > Ph is not well supported from Figure 5, ThF and Th actually show very similar lifetime. Also, in photocatalysis, the system has sacrificial donor and Pt, the charge recombination kinetics should be different from the pure Pdots system. One should not simply compare the charge recombination in a pure Pdots system and apply the conclusion to a photocatalytic system.

Response: We appreciate the valuable comment provided by the reviewer. In Figure 5, our aim was to focus on the intrinsic properties of the polymer structure independent of any external factors in order to explore their effect on charge separation using TAS techniques. The main idea behind our designated polymers is to increase the delocalization between repeated acceptors, as this can potentially enhance the carrier lifetime in the excited state and facilitate efficient charge separation between the donor and acceptor.

We re-measured the TAS data of samples (without Pt or AA) under the same conditions. Upon re-measuring the TAS, we noticed a change in the decay lifetime of PITIC-ThF Pdot. The decay lifetime showed consistency with different linkers, as depicted in the modified **Figure 5**.

Furthermore, the different linkers resulted in varying dihedral angle values between the linker and the acceptor. Our findings indicate that the Ph linker exhibited a larger dihedral angle (34°) compared to the Th linker (23°) and the ThF linker (18°). This observation of a reduced dihedral angle aligns with an increase in HER efficiency and the decay lifetime, which can be attributed to enhanced delocalization between repeated acceptors.

To further validate our hypothesis, we performed steady-state PL measurements on samples with different linkers. As a result, our findings demonstrate a significant difference in the PL quenching (**Figure S36a**) and TA decay lifetime of the Ph, Th, and ThF linkers (Figure 5), which corresponds to their respective dihedral angle values. This discovery has been incorporated into the revised manuscript's Figure 5.

Figure 5. Transient absorption spectra of (a) PITIC-Ph Pdts (water solution), (b) PITIC-Th Pdts, and (c) PITIC-ThF Pdts at different delay times. Transient absorption traces of (d) PITIC-Ph Pdot (probed at 568 nm), (e) PITIC-Th Pdts (probed at 732 nm), and (f) PITIC-ThF Pdts (probed at 739 nm). The data were obtained using an excitation wavelength of 550 nm for PITIC-Ph Pdot and 650 for both PITIC-Th and PITIC-ThF Pdot, using a power of 10 μW . (g) Schematic diagram presenting the effect of different linkers on the charge transfer between acceptors of repeated moieties of the polymer and the charge recombination between the acceptor and donor.

Figure S36a: Photoluminescence emission spectra of PITIC-X Pdots.

We concur with the reviewer regarding the difference between the pure Pdot system and the full photocatalytic system, specifically with regard to the influence of sacrificial and Pt cocatalysts on the photocatalytic reaction. Moreover, according to the reviewer’s suggestion we study the effect of ascorbic acid (AA) as a sacrificial reagent and Pt as a cocatalyst on PITIC-ThF Pdot and PITIC-Th Pdot using the TA spectroscopy and the data presented in the following figures. The following discussion and figures are incorporated into the revised manuscript and SI as the following.

“Durrant & McCulloch have reported that the kinetics of charge recombination in the pure Pdot system differ from those in the full photocatalytic system, where the presence of a scavenger and Pt cocatalyst has a significant impact.⁸ To gain a better understanding of the effect of Pt and scavenger on the photocatalytic activity, transient absorption spectroscopy (TAS) was employed. The TAS spectra of neat PITIC-ThF Pdot in the wavelength region (600–980 nm) were recorded after excitation at 650 nm, and the results were analyzed (**Figure 6a**). The TAS spectra of PITIC-ThF Pdot show a broad ground state bleaching (GSB) between 600 nm and 815 nm, with a maximum peak at 750 nm. Additionally, a photoinduced absorption (PIA) was observed starting from 815 nm and reaching a maximum in the NIR region at 959 nm. This PIA is attributed to the singlet exciton absorption of PITIC-ThF Pdot. In **Figure 6b**, the TAS spectra reveals a larger amplitude with the addition of 3% Pt to the Pdot solution, which is consistent with suppressed bimolecular recombination due to electron transfer to Pt. On the other hand, the addition of AA for Pdot samples with Pt strongly reduced the amplitude in **Figures 6c**, indicating efficient hole extraction in the photocatalytic system.⁸ The exciton decay dynamics of neat PITIC-ThF Pdot and the photocatalytic system containing Pt or Pt+AA were compared in **Figure 6e**. It was found that the addition of Pt to PITIC-ThF Pdot resulted in a longer-lived decay transient compared to the neat PITIC-ThF Pdot, which is consistent with slower bimolecular recombination kinetics with Pt. The further addition of AA resulted in an accelerated decay of the PITIC-ThF Pdot absorption, consistent with hole transfer to AA.^{9, 10} This acceleration was increased even more with AA alone, indicating that the presence of AA scavenger in this photocatalytic system is crucial for achieving high photocatalytic hydrogen evolution rate (**Figure S38**). The same behavior was noticed with PITIC-Th Pdot in the presence of Pt or AA+Pt as presented in **figure S39**. These results provide unequivocal evidence that

the presence of AA retards the recombination of photo-generated charge carriers, leading to a nearly three-fold increase in recombination time in the presence of AA and PITIC-ThF Pdot. These results provide insight into the mechanisms of the photocatalytic system and could be useful for the design of more efficient photocatalysts.

Figure 6: Transient absorption spectra for various samples, including pure PITIC-ThF Pdot (a), PITIC-ThF Pdot with 3 wt% Pt (b), and PITIC-ThF Pdot with 0.1M AA and 3 wt% Pt (c), at different time delays after excitation at 650 nm with a power of 0.9 μW . The decay dynamics of the transient absorption were compared for neat PITIC-ThF Pdot, PITIC-ThF Pdot + Pt, and PITIC-ThF Pdot + AA + Pt, when excited at 650 nm and probed at 959 nm, which were assigned to PITIC-ThF Pdot exciton decay (d).

Figure S38: Transient absorption spectra for PITIC-ThF Pdot with 0.1M AA at different time delays after excitation at 650 nm with a power of 0.9 μ W.

Figure S39: Transient absorption spectra for various samples, including pure PITIC-Th Pdot (a), PITIC-

Th Pdot with 3 wt% Pt (b), and PITIC-Th Pdot with 0.1M AA and 3 wt% Pt (c), at different time delays after excitation at 650 nm with a power of 0.9 μ W. The decay dynamics of the transient absorption were compared for neat PITIC-Th Pdot, PITIC-Th Pdot + Pt, and PITIC-Th Pdot + AA + Pt, when excited at 650 nm and probed at 964 nm, which were assigned to PITIC-Th Pdot exciton decay (d).

The TAS spectra of PITIC-Th Pdot show a broad ground state bleaching (GSB) between 600 nm and 810 nm, with a maximum peak at 740 nm. Additionally, a photoinduced absorption (PIA) was observed starting from 810 nm and reaching a maximum in the NIR region at 964 nm. This PIA is attributed to the singlet exciton absorption of PITIC-ThF Pdot (**Figure S39a**). In **Figure S39b**, the TAS spectra reveals a larger amplitude with the addition of 3% Pt to the Pdot solution, which is consistent with suppressed bimolecular recombination due to electron transfer to Pt. On the other hand, the addition of AA for Pdot samples with Pt strongly reduced the amplitude in **Figures S39c**, indicating efficient hole extraction in the photocatalytic system. The exciton decay dynamics of neat PITIC-ThF Pdot and the photocatalytic system containing Pt or Pt+AA were compared in **Figure S39e**. It was found that the addition of Pt to PITIC-ThF Pdot resulted in a longer-lived decay transient compared to the neat PITIC-ThF Pdot, which is consistent with slower bimolecular recombination kinetics with Pt. The further addition of AA resulted in an accelerated decay of the PITIC-ThF Pdot absorption, consistent with hole transfer to AA.

4. It is not good to normalize the amplitude of TAS data and then only compare the kinetics. Actually, Ph showed much large amplitude that means the system generates more excitons than other polymers (if the absorbance is the same).

Response: We thank the reviewer for his comment, in the text we added a few sentences to clarify the reviewer 'comment as well as we added two more figures to the SI file. It is well known to use the normalized kinetics traces to compare the recombination processes in similar systems which is the same case in our study. Another important point here is that the studied Pdots have different absorption wavelengths and different absorptivity (see **figure S40** in the SI) that is why the different Pdot have different amplitude in the TA. In order to avoid having more than one exciton/Pdot we used low photon flux similar to our previous work on quantum dots-based systems (Nano Lett. 2012, 12, 6393–6399; J. Am. Chem. Soc. 2012, 134, 12110–12117). In our TA measurements, we used 650 nm to pump PITIC-ThF and PITIC-Th while we used 550 nm to pump PITIC-Ph (figure S40).

In the main text, we add the following in the experimental section.

“To avoid non-linear exciton recombination, we used low photon flux to excite one exciton/pdot similar to different quantum dots-based systems. We used two different wavelengths to excite the studied systems 640 nm to pump PITIC-ThF and PITIC-Th and 550 nm to pump PITIC-Ph (**Figure S40**).”

Figure S40: Absorptivity of PITIC-X polymers with different linkers (Ph, Th, and ThF) measured as Pdot in aqueous solution.

5. Prepare polymer particles without surfactant is well-known. The author should not claim it as “our method”

Response: We thank the reviewer for his valuable notes. We understand that the method for preparing polymer particles without surfactant is well-known. Per the reviewer’s suggestion, we have modified the text to make it clear that we are referring to a modified version of the known method.

6. Figure 6b, the comparison is not informative. It is good to compare the same Pdots with or without Pt.

Response: We thank the reviewer for his valuable comment. We agree with the reviewer and in response to the reviewer's feedback, we dispensed **figure 6b** and conducted transient absorption spectroscopy measurements to compare the performance of our top-performing photocatalyst, the PITIC-ThF Pdot, with and without a Pt cocatalyst. As previously discussed, (in Comment 3), our goal was to investigate the impact of Pt on the charge recombination of the PITIC-ThF Pdot. In **Figure 6b**, the results of our TAS measurements show a larger amplitude when 3% Pt was added to the Pdot solution, indicating that electron transfer to Pt may have suppressed bimolecular recombination. Additionally, we found that the addition of Pt to the PITIC-ThF Pdot resulted in a longer decay transient compared to the neat PITIC-ThF Pdot, suggesting slower bimolecular recombination kinetics in the presence of Pt. All figures and discussion have been added to the manuscript.

Figure 6: Transient absorption spectra for various samples, including pure PITIC-ThF Pdot (a), PITIC-ThF Pdot with 3 wt% Pt (b), and PITIC-ThF Pdot with 0.1M AA and 3 wt% Pt (c), at different time delays after excitation at 650 nm with a power of 0.9 μW . The decay dynamics of the transient absorption were compared for neat PITIC-ThF Pdot, PITIC-ThF Pdot + Pt, and PITIC-ThF Pdot + AA + Pt, when excited at 650 nm and probed at 959 nm, which were assigned to PITIC-ThF Pdot exciton decay (d).

7. Why not to study the reductive quenching from ascorbic acid to Pdots? If the author want to provide more information on charge recombination as stated in the title.

Response: We would like to express our gratitude to the reviewer for their insightful comment. As per the reviewer's suggestion, we conducted an investigation on the effect of ascorbic acid (AA) on the charge recombination of the PITIC-ThF Pdot photocatalyst. As illustrated in the response to Comment 3, the addition of AA to both the Pdot samples with and without Pt resulted in a significant reduction in the amplitude, as shown in **Figures 6c and 6d**, respectively. This observation indicates that efficient hole extraction was achieved in the photocatalytic system. Furthermore, we found that the addition of AA in the presence of Pt accelerated the decay of the PITIC-ThF Pdot absorption, consistent with hole transfer to AA. This acceleration was further enhanced with AA alone (**Figure S38**), which suggests that the presence of AA scavenger in the photocatalytic system is crucial for achieving a high photocatalytic hydrogen evolution rate. All figures and discussion have been added to the manuscript.

Figure 6: Transient absorption spectra for various samples, including pure PITIC-ThF Pdot (a), PITIC-ThF Pdot with 3 wt% Pt (b), and PITIC-ThF Pdot with 0.1M AA and 3 wt% Pt (c), at different time delays after excitation at 650 nm with a power of 0.9 μW . The decay dynamics of the transient absorption were compared for neat PITIC-ThF Pdot, PITIC-ThF Pdot + Pt, and PITIC-ThF Pdot + AA + Pt, when excited at 650 nm and probed at 959 nm, which were assigned to PITIC-ThF Pdot exciton decay (d).

Figure S38: Transient absorption spectra for PITIC-ThF Pdot with 0.1M AA at different time delays after excitation at 650 nm with a power of 0.9 μ W.

8. Why did the author use different Pdots concentrations for visible and NIR experiments? No explanations found. The Pdots concentration needs to be provided in different experiments.

Response: We appreciate the valuable feedback provided by the reviewer. In Figure 3c, we optimized the weight of the PITIC-ThF Pdot photocatalyst for visible light HER by testing different weights (0.1, 1, 2, 3, and 5 mg/10 mL). After optimization, we proceeded to test the HER under NIR irradiation using only one weight (5mg/10mL) of the optimized photocatalyst. In order to provide clarity to the reader, we will revise our description of the photocatalytic H₂ evolution experiment in the manuscript accordingly.

“For measuring the H₂ evolution under visible light, the Pdot solution prepared with different weights (0.1, 1, 2, 3, and 5 mg/10 mL) containing Ascorbic Acid (AA (0.1 M)) and 3% H₂PtCl₆ cocatalyst was inserted into the reaction glass container and sealed tightly with a septum.”

Moreover, we performed two separate tests on the HER of our best material (PITIC-ThF Pdot) using visible and NIR light, respectively, while utilizing the same Xenon lamp as the light source. For both experiments, we dispersed 5.0 mg of PITIC-ThF Pdot in 10 mL of 0.1M AA solution with 3% Pt as the cocatalyst. A light cut filter at $\lambda > 380$ nm and a light intensity of 1000 W/m² were used in the visible light experiment, while in the NIR experiment, a light cut filter at $\lambda > 780$ nm and a light intensity of 3000W/m² were employed. In Figure S26a, PITIC-ThF Pdot achieved a HER of 171.3 μ mol/h under visible light and 20.2 μ mol/h under NIR light using the Xenon lamp as the light source.

Figure S26: (a) Time course of the produced H_2 for the PITIC-ThF under NIR light and Visible light using the same light source (Xe lamp), (b) Comparison between the Xe lamp and LED lamp as different sources of the visible light for H_2 production using PITIC-ThF photocatalyst. We performed two separate tests on the HER of our best material (PITIC-ThF Pdot) using visible and NIR light, respectively, while utilizing the same Xenon lamp as the light source. For both experiments, we dispersed 5.0 mg of PITIC-ThF Pdot in 10 mL of 0.1M AA solution with 3% Pt as the cocatalyst. A light cut filter at $\lambda > 380$ nm and a light intensity of 1000 W/m^2 were used in the visible light experiment, while in the NIR experiment, a light cut filter at $\lambda > 780$ nm and a light intensity of 3000 W/m^2 were employed. In Figure S26a, PITIC-ThF Pdot achieved a HER of $171.3 \mu\text{mol/h}$ under visible light and $20.2 \mu\text{mol/h}$ under NIR light using the Xenon lamp as the light source.

9. The author need to well proof-read the manuscript, there are many grammar errors and typos.

Response: We are grateful for the reviewer's suggestion. Per the reviewer's suggestion, we have thoroughly proof-read the manuscript to identify and correct the errors, including grammar and typos.

Again, we are highly grateful to all the reviewers for their time and constructive input. We hope that our responses would help to clarify certain issues and, of course, in making the revised manuscript suitable for publication. Last, we thank you for your time and consideration and we look forward to hearing from you soon.

Sincerely,

Ho-Hsiu Chou, Ph.D.
 Associate Professor
 Department of Chemical Engineering
 National Tsing Hua University

REVIEWER COMMENTS

Reviewer #2 (Remarks to the Author):

The authors have well addressed the comments from the reviewer. One thing I could not find in this work is the polymer weight which is important to claim the material is polymer and make the work reproducible. I suggest that the authors provide GPC data of the new polymer.

Reviewer #3 (Remarks to the Author):

This study demonstrates a polymer acceptor-based photocatalytic system that exhibits impressive NIR-responsive HER activity. Although similar systems have been reported recently, this work attempts to put forward a new point of view from the perspective of charge recombination. Before the article is considered for possible acceptance, I think the following points need clarification:

Comment 1: As the authors emphasize, the photophysical properties of a polymer strongly depend on its structural features, including dihedral angles and interatomic spacing within the structure. However, the dihedral angles between polymer segment units strongly depend on its packing structure, which cannot be obtained simply by performing DFT or any kinetic optimization on a single segment. If possible, try to provide more information about the internal chain segment packing of the nanocrystal (high crystallinity claimed by the author) to obtain accurate structural information such as "dihedral angle".

Comment 2: It should be pointed out that the calculation of B3LYP/6-31G* level seems to be completely unable to describe the dispersion effect due to the wrong long-range behavior of the correlation potential, which is not conducive to the optimization of long-chain macromolecules. The most effective way to solve the poor ability of these functionals to describe the dispersion effect is to introduce empirical dispersion correction terms. While more calculation details need to be given, I don't even seem to find the author to use the program.

Comment 3: The authors show isosurfaces of several frontier orbitals in Figures S30-S34, however, this does not mean that these electronic transitions are easy to occur. Current evidence lacks an understanding of the excited states involved. The authors may consider using time-dependent density functional or other methods for molecular excitation calculations.

Comment 4: The characterization and understanding of the co-catalyst structure in the manuscript is still insufficient. Pt may be supported on Pdots in the form of particles, or it may exist in the system in the form of single atoms, oxides, Pd-Pt complexes, or even free clusters (charge transfer through collisions). The analysis of Pt-related electronic excitations under this ambiguity is incomplete.

Comment 5: In Fig. 5d-f, the authors consider the relaxation lifetime of negative features in TAS to measure the carrier recombination time. However, there is not sufficient evidence that the bleach recovery lifetime is related to the charge recombination here. Further data are needed for peak assignment.

Comment 6: The test details of transient/steady-state spectroscopy data need to be provided as much as possible, which is very important for the confirmation and comparison of related mechanisms. In particular, the concentration of the sample dispersed in the TAS, the power of the laser and other information.

Comment 7: The transient absorption spectra involved in pure PITIC-ThF Pdot, PITIC-ThF Pdot with 3 wt% Pt, and PITIC-ThF Pdot with 0.1M AA and 3 wt% Pt in Figure 6 can be considered to increase the

number of averages to improve data quality. The current signal-to-noise ratio (especially the early stage) is difficult to accurately measure the decay dynamics, and it is easy to obtain wrong fitting and kinetic trends.

Response to the reviewers' comments

We express our gratitude to the reviewers for their thorough review of our manuscript. We have carefully considered their comments, which we deem to be highly relevant and significant, and have made appropriate changes to the paper in response.

Our point-by-point response to the reviewers' comments (highlighted in **blue**) and their comments and remarks (in **black**) is presented below, along with a detailed explanation of the corresponding revisions and additions that we have incorporated to adequately address them in the revised manuscript and Supplementary Information.

Reviewer #2 (**Remarks** to the Author):

The authors have well addressed the comments from the reviewer. One thing I could not find in this work is the polymer weight which is important to claim the material is polymer and make the work reproducible. I suggest that the authors provide GPC data of the new polymer.

Response: We thank the reviewer for his positive recommendation of our work. According to the reviewer suggestion, we determine the polymer weight using the GPC data as presented in the following table.

Table S4: The determined weight of all polymers from GPC data.

Polymer blends	Mw	Mn	PDI (Mw / Mn)
PITIC-Ph	9900	3550	2.79
PITIC-Th	10180	6320	1.61
PITIC-ThF	8950	4320	2.07
PBTIC-Ph	12430	6340	1.96
PBTIC-Th	9270	3610	2.57
PBTIC-ThF	10430	5930	1.75

Responses to Reviewer #3

This study demonstrates a polymer acceptor-based photocatalytic system that exhibits impressive NIR-responsive HER activity. Although similar systems have been reported recently, this work attempts to put forward a new point of view from the perspective of charge recombination. Before the article is considered for possible acceptance, I think the following points need clarification:

Comment 1: As the authors emphasize, the photophysical properties of a polymer strongly depend on its structural features, including dihedral angles and interatomic spacing within the structure. However, the dihedral angles between polymer segment units strongly depend on its packing structure, which cannot be obtained simply by performing DFT or any kinetic optimization on a single segment. If possible, try to provide more information about the internal chain segment packing of the nanocrystal (high crystallinity claimed by the author) to obtain accurate structural information such as "dihedral angle".

Response: We thank the reviewer for his valuable comment. To more understand the role of the dihedral angle in dominating the photophysical property, structural information about its influence on the internal chain segmental packing in PBTIC-Th Pdot and PBTIC-ThF Pdot (taken as model examples) was further examined by the grazing-incidence wide-angle X-ray scattering (GIWAXS), as shown in Figure S34. 2D GIWAXS patterns of thin films of the two samples spin-coated on Si wafers displayed isotropic scattering (**Figure S34a**), revealing that the chain segmental packing did not have preferred orientation. As confirmed by the corresponding 1D GIWAXS data, the in-plane and out-of-plane scattering intensity profiles of each sample exhibited almost the same positions of the diffraction peaks (**Figure S34b vs**

Figure S34c). However, in comparison to the case of PBTIC-Th Pdot, the peak positions observed for PBTIC-ThF Pdot were located at higher q and the diffraction peak at $q = 2.53 \text{ \AA}^{-1}$ characteristic of the chain segmental packing became sharper. This observation indicated that the substitution with fluorines to form stronger hydrogen bonds for reducing dihedral angle into the smaller one (i.e., changing from 23.41 to 17.66°) effectively improved the ordering along with a reduction in the average spacing of the internal chain segmental packing. That is to say, the improved photophysical properties of PBTIC-ThF Pdot were very likely subjected to the development of a more ordered nanocrystal, mainly endowed with a smaller dihedral angle. This part was described in supporting information **Figure S34**.

Figure S34. (a) 2D GIWAXS patterns measured for PBTIC-Th Pdot and PBTIC-ThF Pdot thin films spin-coated on Si wafers. The corresponding 1D GIWAXS profiles, including in-plane and out-of-plane scattering profiles ascribed to the scattering with a direction parallel and perpendicular to the thin film

plane, respectively, are shown in (b) and (c).

As for the theoretical part, to better understand the effect of the dihedral angle on the excited-state properties, we performed dihedral scanning over the dihedral angle between the linker and the neighboring acceptor. The excitation energy profile along the dihedral angle is shown in Figure S33. While the magnitude of vertical excitation energies is slightly different among the polymer series of different linkers, their energy profile along the dihedral angle is quite similar. Moreover, the excitation energy ($E_{S_n} - E_{S_0}$) remains nearly the same upon changing the dihedral, which indicates that the disorder introduced through L-A dihedral angle distribution shall barely alter the excitation energy. A paragraph discussing the electronic structure of these dimers was added to manuscript page no. 16 as follows.

“Finally, the excited-state properties of these polymers were investigated using TD-DFT using the aforementioned dimer model. The vertical S_1 energy remains relatively close within each series, with the polymer containing Ph exhibiting a slightly higher E_{S_1} value. While the computational E_{S_1} trend based on the dimer model agree qualitatively with our experiment, it fails to explain a much larger difference in the optical bandgap (> 0.1 eV) between the Ph-containing and Th/ThF-containing polymers. We assumed that the different linker-acceptor dihedral angle distributions of different polymers may be the reason for this discrepancy. Therefore, we conducted a dihedral angle scanning analysis to compute the excited-state energy profile along the linker-accepter dihedral angle (Figure S33). However, all polymers exhibit very similar characteristics in their energy profiles, which still could not explain the disparities between our computational and experimental findings. We suggest that this distinction may be explained by their different polymer packing behaviors, where the interchain coupling and the dielectric effect can significantly alter the excited-state properties.”

Figure S33. The relative energy levels along the linker-acceptor dihedral angle scanning of ground state (S_0) and excited states (S_1 , S_2 and S_3) for (a) PBTIC-Ph, (b) PBTIC-Th, (c) PBTIC-ThF, (d) PITIC-Ph, (e) PITIC-Th, and (f) PITIC-ThF. The dotted line represents the $k_B T$ at the room temperature.

Comment 2: It should be pointed out that the calculation of B3LYP/6-31G* level seems to be completely unable to describe the dispersion effect due to the wrong long-range behavior of the correlation potential, which is not conducive to the optimization of long-chain macromolecules. The most effective way to solve the poor ability of these functionals to describe the dispersion effect is to introduce empirical dispersion correction terms. While more calculation details need to be given, I don't even seem to find the author to use the program.

Response: We thank the reviewer for pointing this out. To address the issue, we recomputed all computations with a new functional ω B97X-D, which includes a D2 dispersion correction. A detailed computational description was also added in the methodology section:

“Computational details

We constructed an ADA-L-ADA dimer model to simulate the properties of each polymer. All geometry optimizations were performed using DFT at the ω B97X-D/6-31G(d) level of theory. The excited-state computations were performed using TD-DFT with Tamm-Dancoff Approximation at ω B97X-D/6-31G(d) level of theory. The dihedral scanning was performed using a constrained-optimization scheme along the linker-acceptor dihedral defined in **Figure S33**. The $\lambda_{S_1 \rightarrow S_0}$ is computed using the following equation:

$$\lambda_{S_1 \rightarrow S_0} = E_{S_0/S_1} - E_{S_0/S_0}$$

where E_{S_0/S_1} and E_{S_0/S_0} are the S_0 energy computed at the geometry of S_1 minimum and S_0 minimum, respectively. All aforementioned computations were computed using GAUSSIAN16. Finally, the electronic coupling between the two neighboring acceptors connected through a linker (ADA-L-ADA) was computed using CI-CDFT at the ω B97X-D/6-31G(d) level of theory implemented in QChem v5.4.1.”

Repeated theoretical part explained in the manuscript and supporting information as the following:

“we employed density functional theory (DFT) and transient absorption (TA) spectroscopy. To better understand the effect of linker on the properties of each polymer, we performed computations using density functional theory (DFT), configuration-interaction constrained DFT (CI-CDFT) and time-dependent DFT (TD-DFT). For the computational details, please refer to the Methodology section and the supporting information. First, the dihedral angles between the linker and the acceptor decrease as we shift from using Ph, Th, to ThF as the linker, which holds for both PBTIC and PITIC series (as shown in **Figure 4a**). This is because the H-H repulsion present in the benzene linker and the acceptor reduces by replacing it with Th linker. Upon replacing Th with ThF, not only does this substitution decrease H-H repulsion, but it also introduces the F-H attraction, resulting in a further reduction of the dihedral angle (**Figure S30**).^{46, 47, 48} The smaller dihedral angle between the acceptor and π -linker in PITIC-ThF indicates a more planarized structure with enhanced charge carrier mobility and transfer between the acceptors of different repeated moieties. This, in turn, improves the charge separation

between the donor and acceptor for each polymer repeated moiety, resulting in an enhanced exciton dissociation yield and improved photocatalytic activity. Conversely, increasing the dihedral angle with Th and Ph π -linkers reduces planarity and charge transfer between the acceptors, leading to increased charge recombination from the donor to acceptor and consequently less efficient photocatalytic activity (**Figure 4b**). As for electronic structure, all dimers exhibit a nearly degenerate HOMO/HOMO-1 and LUMO/LUMO+1, resulting from the linear combination of HOMOs and LUMOs of the corresponding monomers (as shown in **Figure S31** and **Figure S32**). While all energy values of frontier orbitals remain nearly unchanged across both the PITIC and PBTIC series, it is noteworthy that the molecular orbital distribution extends slightly more into the linker unit in the cases involving Th and ThF for both polymer series. This observation could be partly attributed to the increased planarity found in the polymers containing Th and ThF. To understand how these different structural features affect the electron transport properties, we estimate the electron transfer from an acceptor to another neighboring acceptor. To achieve this, we computed the electronic coupling ($V_{A_1A_2}$) between these two states using CI CDFT, where the electron transfer rate is proportional to $|V_{A_1A_2}|^2$. As shown in **Table 7**, the magnitude of $|V_{A_1A_2}|$ exhibits an ascending trend in the sequence of Ph, Th, and ThF for both series of polymers. Our findings imply that electron transfer along the polymer chain is more favorable when utilizing the ThF linker, which could be attributed to the increased planarity observed in the polymer chain containing ThF. Finally, the excited-state properties of these polymers were investigated using TD-DFT using the aforementioned dimer model. The vertical S_1 energy remains relatively close within each series, with the polymer containing Ph exhibiting a slightly higher E_{S_1} value. While the computational E_{S_1} trend based on the dimer model agree qualitatively with our experiment, it fails to explain a much larger difference in the optical bandgap (> 0.1 eV) between the Ph-containing and Th/ThF-containing polymers. We assumed that the different linker-acceptor dihedral angle distributions of different polymers may be the reason for this discrepancy. Therefore, we conducted a dihedral angle scanning analysis to compute the excited-state energy profile along the linker-accepter dihedral angle (**Figure S33**). However, all polymers exhibit very similar characteristics in their energy profiles, which still could not explain the disparities between our computational and experimental findings. We suggest that

this distinction may be explained by their different polymer packing behaviors, where the interchain coupling and the dielectric effect can significantly alter the excited-state properties (see **Figure S34** and its discussion). Finally, we computed the reorganization energy ($\lambda_{S_1 \rightarrow S_0}$) of the $S_1 \rightarrow S_0$ transition, where organic semiconductors with small $\lambda_{S_1 \rightarrow S_0}$ values are believed to exhibit slower radiationless relaxation and thereby giving higher quantum yield of photoconversion processes. As shown in Table 7, the $\lambda_{S_1 \rightarrow S_0}$ slightly decreases as we shift from using Ph, Th, to ThF as the linker for both polymer series. The trend in $\lambda_{S_1 \rightarrow S_0}$ agrees with the trend of the excited-state lifetime observed in the fs-TAS.

(b) Effect of dihedral angle on the charge transfer

Figure 4. (a) The dihedral angle between the linker and the acceptor (indicated in red color) for the dimer of each polymer. (b) Schematic diagram showing the dihedral angle effect of different π -linkers on the charge transfer between acceptors of repeated moieties.

Figure S31. The frontier orbitals (isovalue: 0.015 \AA^3) of PBTIC series.

Figure S32. The frontier orbitals (isovalue: 0.015 \AA^3) of PITIC series.

Figure S33. The relative energy levels along the linker-acceptor dihedral angle scanning of ground state (S_0) and excited states (S_1 , S_2 and S_3) for (a) PBTIC-Ph, (b) PBTIC-Th, (c) PBTIC-ThF, (d) PITIC-Ph, (e) PITIC-Th, and (f) PITIC-ThF. The dotted line represents the $k_B T$ at the room temperature.

Table S7. The excited-state and charge-transfer properties for each polymer.

	PBTIC-Ph	PBTIC-Th	PBTIC-ThF	PITIC-Ph	PITIC-Th	PITIC-ThF
$ V_{A_1 A_2} ^a$	2.49	5.37	6.08	2.70	5.44	5.82
$E_{S_1}^b$	2.52	2.50	2.50	2.63	2.61	2.61
$\lambda_{S_1 \rightarrow S_0}^c$	131	128	127	158	155	154

^a The electronic coupling element (in meV) of the electron transfer between the neighboring acceptor sites computed using CDFT.

^b The S_1 vertical excitation energy (in eV) computed using TD-DFT.

^c The reorganization energy (in meV) for transition from S_1 to S_0 state computed using DFT and TD-DFT.

Comment 3: The authors show isosurfaces of several frontier orbitals in Figures S30-S34, however, this does not mean that these electronic transitions are easy to occur. Current evidence lacks an understanding of the excited states involved. The authors may consider using time-dependent density functional or other methods for molecular excitation calculations.

Response: We performed excited-state computations using TD-DFT with Tamm-Dancoff Approximation at ω B97X-D/6-31G(d) and the dimer model for all polymers. The excitation energy profile along the L-A dihedral angle was investigated, as discussed in the response to Comment 1. And the associated computational details are also described in the response to Comment 2. In addition, we computed the reorganization energy of $S_1 \rightarrow S_0$ transition ($\lambda_{S_1 \rightarrow S_0}$), which is in good agreement with the trend of excited-state lifetime obtained from our fs-TAS measurement. A paragraph discussing the electronic structure of these dimers was added to manuscript page no. 16 as follows:

“Finally, we computed the reorganization energy ($\lambda_{S_1 \rightarrow S_0}$) of the $S_1 \rightarrow S_0$ transition, where organic semiconductors with small $\lambda_{S_1 \rightarrow S_0}$ values are believed to exhibit slower radiationless relaxation and thereby giving higher quantum yield of photoconversion processes. As shown in Table S7, the $\lambda_{S_1 \rightarrow S_0}$ slightly decreases as we shift from using Ph, Th, to ThF as the linker for both polymer series. The trend in $\lambda_{S_1 \rightarrow S_0}$ agrees with the trend of the excited-state lifetime observed in the fs-TAS.”

Table S7. The excited-state and charge-transfer properties for each polymer.

	PBTIC-Ph	PBTIC-Th	PBTIC-ThF	PITIC-Ph	PITIC-Th	PITIC-ThF
$ V_{A_1 A_2} ^a$	2.49	5.37	6.08	2.70	5.44	5.82
$E_{S_1}^b$	2.52	2.50	2.50	2.63	2.61	2.61
$\lambda_{S_1 \rightarrow S_0}^c$	131	128	127	158	155	154

^a The electronic coupling element (in meV) of the electron transfer between the neighboring acceptor sites computed using CDFT.

^b The S_1 vertical excitation energy (in eV) computed using TD-DFT.

^c The reorganization energy (in meV) for transition from S_1 to S_0 state computed using DFT and TD-DFT.

Comment 4: The characterization and understanding of the co-catalyst structure in the manuscript is still insufficient. Pt may be supported on Pdots in the form of particles, or it may exist in the system in the form of single atoms, oxides, Pd-Pt complexes, or even free clusters (charge transfer through collisions). The analysis of Pt-related electronic excitations under this ambiguity is incomplete.

Response: We want to express our appreciation to the reviewer for their invaluable feedback. The question raised by the reviewer, particularly regarding the precursor of Pt employed in our photocatalytic system (H_2PtCl_6), is of considerable interest. In this research, our primary focus was on the discussion and comparison of the performance of various polymer structures and linkers under identical photocatalytic conditions. It's worth mentioning that prior researchers and our research team have previously examined aspects such as the quantity of co-catalysts or the oxidation state of Pt in earlier publications [Adv. Mater. 2021, 34, 2105007; Angew. Chem. Int. Ed. 2023, 62, e202217989; ACS Appl. Mater. Interfaces 2021, 13, 56554-56565; ACS Appl. Energy Mater. 2022, 5, 7950-7955]. Therefore, in this study, we introduced Pt using the same procedure to ensure a uniform testing environment for each polymer, allowing for meaningful comparisons. Consequently, the specific type of Pt was not further investigated in this study, as it fell outside the scope of our intended discussion and investigation.

Comment 5: In Fig. 5d–f, the authors consider the relaxation lifetime of negative features in TAS to measure the carrier recombination time. However, there is not sufficient evidence that the bleach recovery lifetime is related to the charge recombination here. Further data are needed for peak assignment.

Response: We extend our appreciation to the reviewer for sharing their comment. We must respectfully express our disagreement with their argument. It is widely acknowledged that the relaxation lifetime of bleach recovery is intricately linked to charge injection/recombination in numerous materials, particularly those with semi-conductive properties, such as quantum dots. This has been substantiated by numerous prior investigations, as evidenced by the references below:^{3, 4}

3. Židek K, *et al.* Electron Transfer in Quantum-Dot-Sensitized ZnO Nanowires: Ultrafast Time-Resolved Absorption and Terahertz Study. *J. Am. Chem. Soc.* **134**, 12110-12117 (2012).
4. Židek K, Zheng K, Abdellah M, Lenngren N, Chábera P, Pullerits T. Ultrafast Dynamics of Multiple Exciton Harvesting in the CdSe–ZnO System: Electron Injection versus Auger Recombination. *Nano Letters* **12**, 6393-6399 (2012).

Furthermore, the photoluminescence spectrum (as shown in Figure S35) provides compelling evidence of variations in charge recombination associated with different linkers. The results are notably consistent with the relaxation lifetime observed in the transient absorption spectrum.

Figure S35: Steady-state photoluminescence (PL) spectra of all presented polymers nanoparticles in water.

Comment 6: The test details of transient/steady-state spectroscopy data need to be provided as much as possible, which is very important for the confirmation and comparison of related mechanisms. In particular, the concentration of the sample dispersed in the TAS, the power of the laser and other information.

Response: We appreciate the reviewer’s note. The details of the transient absorption spectroscopy

measurements are provided as the following.

“Time-resolved experiments were performed using laser-based spectroscopy, with a laser power of less than one photon absorption per particle. Samples for transient absorption experiments were prepared as polymer nanoparticles solution (1 mg /10 mL) and kept in the dark between measurements. A Coherent Legend Ti: Sapphire amplifier (800 nm, 100 fs pulse length, 3 kHz repetition rate) was used. The output was split to pump and probe beams. Excitation pulses at specific wavelengths were acquired using an optical parametric amplifier (Topas C, Light Conversion). The probe pulses (a broad supercontinuum spectrum) were generated from the 800-nm pulses in a CaF₂ crystal and split by a beam splitter into a probe pulse and a reference pulse. The probe pulse and the reference pulse were dispersed in a spectrograph and detected by a diode array. The instrumental response time was approximately 100 fs. The kinetic traces were fitted with a sum of convoluted exponentials:

$$Y(t) = \text{ext} \left[-\frac{(t-t_0)^2}{\tau_p} \right] * \sum_i A_i \exp \left(-\frac{t-t_0}{\tau_i} \right)$$

where $\tau_p = \frac{\text{IRF}}{2 \ln 2}$ and IRF is the width of the instrument response function (full width at half-maximum), t_0 is the time zero, A_i and τ_i are amplitude and decay times, respectively, and * is the convolution operator.

To avoid non-linear exciton recombination, we used low photon flux to excite one exciton/Pdot similar to different quantum dots-based systems. We used two different wavelengths to excite the studied systems 650 nm to pump PITIC-ThF and PITIC-Th and 550 nm to pump PITIC-Ph (figure S40).”

Comment 7: The transient absorption spectra involved in pure PITIC-ThF Pdot, PITIC-ThF Pdot with 3 wt% Pt, and PITIC-ThF Pdot with 0.1M AA and 3 wt% Pt in Figure 6 can be considered to increase the number of averages to improve data quality. The current signal-to-noise ratio (especially the early stage) is difficult to accurately measure the decay dynamics, and it is easy to obtain wrong fitting and kinetic trends.

Response: We appreciate the valuable feedback provided by the reviewer. We agree with the reviewer's comments for suggesting increasing the number of averages to improve data quality, and indeed, in

Figure 6 the transient absorption spectroscopy (TAS) measurements were conducted with averaging the results from >10 scans, but with very low power energy (0.9 μW). Our choice of low-power energy settings was aimed at minimizing non-linear effects.

Per the reviewer's suggestion, to validate our data trends while improving the signal-to-noise ratio, we conducted additional measurements using higher power energy (10 μW). These supplementary measurements clearly demonstrate that we were able to obtain the same kinetics data trends for different linkers with a more favorable signal-to-noise ratio. The part showing the additional measurements using higher power energy (10 μW) was added to supplementary information Figure S39 as follows:

Figure S39: Transient absorption spectra for various samples, including pure PITIC-ThF Pdot (a), PITIC-ThF Pdot with 0.1M AA (b), PITIC-ThF Pdot with 0.1M AA and 3 wt% Pt (c), and PITIC-ThF Pdot with 3 wt% Pt, at different time delays after excitation at 650 nm with a power of 10 μW . The decay dynamics of the transient absorption were compared for neat PITIC-ThF Pdot, PITIC-ThF Pdot + Pt, PITIC-ThF Pdot + AA, and PITIC-ThF Pdot + AA + Pt, when excited at 650 nm and probed at

959 nm, which were assigned to PITIC-ThF Pdot exciton decay (d).

Sincerely,

Ho-Hsiu Chou, Ph.D.

Associate Professor

Department of Chemical Engineering

National Tsing Hua University

REVIEWERS' COMMENTS

Reviewer #3 (Remarks to the Author):

After revised, this manuscript has clarified questions, and it can be accepted as current form.